# Interannual Variations in Siberian Carbon Uptake and Carbon Release Period.

Dieu Anh Tran[1], Christoph Gerbig[1], Christian Rödenbeck[1], Sönke Zaehle[1].

[1]Department of Biogeochemical Signal, Max Planck Institute for Biogeochemistry, Jena, 07745, Germany.

*Correspondence to*: Dieu Anh Tran (atran@bgc-jena.mpg.de)

**Abstract.** Winters with higher-than-average temperatures are expected to enhance the respiratory release of $CO_2$, thereby weakening the annual net terrestrial carbon sink. Using the 2010-2021 atmospheric $CO_2$ record from the ZOtino Tall Tower Observatory (ZOTTO) located at 60°48′ N, 89°21′ E, this study analyses inter-annual changes in the timing and intensity of the carbon uptake and release periods (CUP and CRP, respectively) over central Siberia. We complement our $CO_2$ mole fraction analysis with the atmospheric inversion results to disentangle the effects of meteorological variability from the ecosystem's response to climate variability at a regional scale. From the observational data, CRP length and amplitude significantly increased between 2010 and 2021. Similarly, CUP length and amplitude showed a positive but weaker trend since 2010, suggesting increased $CO_2$ release during cold months offset the uptake during the growing season. This suggests that during 2010-2021, climate warming did not lead to higher annual net $CO_2$ uptake despite the enhanced growing season uptake because cold season respiration has also increased due to warming. The observational analysis further showed the influence of two extreme events: the 2012 wildfire and the 2020 heat wave. However, analysis of the inversion-derived net ecosystem exchange flux for the ZOTTO region did not reveal these trends or extreme events. Therefore, while ZOTTO data contains substantial information on the magnitude of the Siberian carbon balance (without further data from additional stations), we could not attribute a distinct contribution of ecosystems in the ZOTTO region of influence to the observed trends and extremes.

## 1 Introduction

Siberian ecosystems play an important role in the global carbon budget. Whether they function as a future net carbon sink or source depends on seasonal climate variability and environmental change (Huemmrich et al., 2010; McGuire et al., 2012; Schuur et al., 2015). High-latitude ecosystems are generally temperature or radiation-limited, and therefore, warming is the main control on the biogeochemistry and bio-geophysics of high-latitude ecosystems and their associated feedback to regional

and global climate (Box et al., 2019; Koven et al., 2011). On the one hand, climate warming promotes a reduction in spring snow cover, an earlier landscape thawing, an earlier onset of vegetation productivity, and longer growing seasons with increased vegetation productivity (Box et al., 2019). Climate warming has thereby contributed to high-latitude greening that has substantially enhanced photosynthetic carbon dioxide ($CO_2$) uptake in the Northern Hemisphere over the past five decades (Ciais et al., 2019). On the other hand, warming-induced early growing season productivity can also increase cumulative

evapotranspiration demand, which can reduce soil moisture levels and increase drought stress (Barnett et al., 2005; Buermann

et al., 2013; Parida & Buermann, 2014; Yi et al., 2014). Recent satellite observations over northern ecosystems have shown widespread moisture stress-induced decline in late growing season productivity, potentially offsetting productivity gains from warmer springs (Buermann et al., 2018), yet there is large uncertainty in the spatial pattern and magnitude of such seasonal compensations (Richardson et al., 2010). During the cold winter months, soil respiration is one of the key processes responsible for the variations in atmospheric $CO_2$. Winters with higher temperatures are expected to increase microbial respirations, enhancing the respiratory release of $CO_2$ and thereby weakening the annual net terrestrial carbon sink (Commane et al., 2017). How fast and strong these feedback processes (i.e., growing season uptake and cold season respiration) operate and which will dominate is still an open and highly pressing scientific question.

Most current studies examining the duration of the carbon uptake period of high-latitude growing seasons primarily utilise flux measurements from eddy covariance (EC) combined with global vegetation models or data-based estimates (atmospheric inversions or satellite-derived data such as the Normalized Difference Vegetation Index (NDVI) (e.g., Gu et al., 2022; Tang et al., 2021; Welp et al., 2016). However, one disadvantage of EC flux measurements is the occurrence of data gaps due to technical failures and challenges in continuing measurements in severe winter conditions, in addition to the need for filtering data collected in low turbulence conditions (especially in the winter), compromising the data coverage of the EC technique. Gap-filling approaches for eddy covariance data exist, but they are associated with elevated uncertainties and may even lead to systematic biases. For example, a recent study showed that a widely used eddy covariance gap-filling method can cause systematic biases, leading to further uncertainties in carbon balance estimates (Vekuri et al., 2023). Furthermore, reliable long-term EC flux data (more than ten years) are currently sparse over arctic and boreal regions. Additionally, EC flux measurements have a local footprint, whereas mole fraction data integrates the signal over a large area, being more representative of a regional spatial scale (Gloor et al., 2001).

Long-term atmospheric $CO_2$ mole fraction observations are an alternative, reliable data source that has been used in numerous studies (e.g., Keeling et al., 1996; Pearman and Hyson, 1981; Bacastow et al., 1985; Myneni et al., 1997; Graven et al., 2017; Piao et al., 2008; Angert et al., 2005) to monitor the dynamics of carbon exchange in northern ecosystems. Most of these studies have concentrated on the amplitude of the seasonal cycle or the spring and summer boundaries of the growing season (Randerson et al., 1999; Piao et al., 2008; Piao et al., 2017), while less attention has been given to winter respiration. Importantly, the spatial scope of these studies predominantly focuses on the Northern American region where two long-term measurement stations are located (Barrow Atmospheric Baseline Observatory (71°29′ N, 156°61′ W) and Alert (82°50′ N, 62°50′ W)), but few studies cover observations from Siberia. Indeed, despite the large significance of the Siberian domain as a climate "hot spot" for carbon storage in the global carbon cycle and its sensitivity to global warming, the Siberian region is only sparsely covered by continuous measurement stations, representative of changes on large spatial scales and decadal time scales. The global observation networks contain, at present, only very few stations for continuous monitoring of the full suite of greenhouse gases in the entire Siberian region north of 45°N (https://cosima.nceas.ucsb.edu/carbon-flux-sites/), and most

of them have now become inaccessible. In the framework of the project "Observing and Understanding Biogeochemical Responses to Rapid Climate Changes in Eurasia", a scientific platform, the Zotino Tall Tower Observation (ZOTTO) facility, was constructed in central Siberia in 2006. Since 2009, at this site, continuous measurements of $CO_2$, $CH_4$ and a suite of additional atmospheric gases, as well as measurements of their isotopic composition, have been performed on a routine basis (Winderlich et al., 2010). Complemented by additional measurements of meteorology, chemically active trace gases, and aerosols, ZOTTO is a continental long-term atmospheric monitoring station which documents and helps to quantify the anticipated changes in biogeochemical cycling in this important but observation-poor region of the globe.

Here, we utilise ZOTTO long-term continuous atmospheric $CO_2$ measurements from 2010-2021 to investigate the interannual variability of the seasonal cycle of $CO_2$ exchange of high-latitude Siberian ecosystems. First, we will assess the quality of the continuous $CO_2$ mole fraction dataset at ZOTTO. We then quantify interannual changes in the timing (i.e., onset and termination) and intensity (i.e., amplitude and length) of the Carbon Uptake Period (CUP) and Carbon Release Period (CRP) and analyse their correspondence with climate anomalies. We finally complement our analysis with the results of an atmospheric inversion to disentangle the effects of meteorological variability in atmospheric tracer transport from ecosystem responses to climate variations.

## 2 Methods

### 2.1 Data

The mole fraction of atmospheric $CO_2$ has been measured at the Zotino Tall Tower Observatory (ZOTTO) located in the middle-taiga subzone (Yenisei region) of Western Siberia on the left bank of the Yenisei River (60°48′ N, 89°2′ E, 114 m above sea level), as described originally in Winderlich et al. (2010). The continuous monitoring of $CO_2$ in the atmospheric surface layer has been conducted since May 2009. Data from the EnviroSense 3000i gas analyser (Picarro Inc., USA) and the set of measuring equipment (including air intakes) situated at the metal mast at the height of 4, 52, 92, 156, 227, and 301 m were calibrated. The calibration system consists of four horizontally stored aluminium tanks (X2019 scale) (Table A1). To monitor the accuracy of the $CO_2$ measurements at ZOTTO, one target tank has been measured every 200 h for 8 min, randomly distributed between two calibration cycles. This data is processed like the ambient air measurement data. After applying the calibration procedure as in Winderlich et al. (2010), the $CO_2$ mole fraction of the target tank was found to be $404.64 \pm 0.04$ ppm for the entire period (Fig. 1). A comparison with target tank values from the Jena GASLAB ($404.60 \pm 0.09$ ppm) indicates a small, but statistically insignificant bias in the observations and no discernible long-term trend in the measurements.

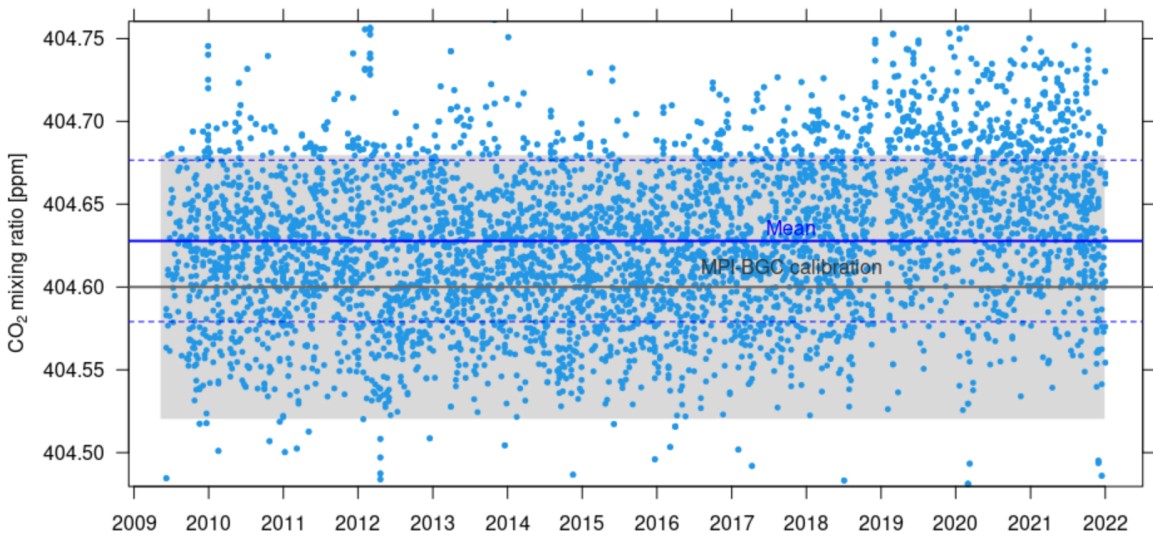

**Figure 1. Target tank time series (blue solid and dashed lines represent the mean ± standard deviation, solid grey line and grey shading is the Jena GASLAB standard ± error).**

To further establish the reliability of the continuous measurements, the measurements are compared to laboratory analyses of

95 flask samples taken approximately weekly at 300 m height (Heimann et al., 2022). To reduce the mismatch between the timing and averaging periods of continuous and flask measurements, we employed the deconvolution approach described exemplarily for one flask measurement by Winderlich et al. (2010). The mean difference ± standard deviation between the in-situ approximation and all currently available flask data is $0.086 \pm 0.32$ ppm for $CO_2$ (Fig. 2).

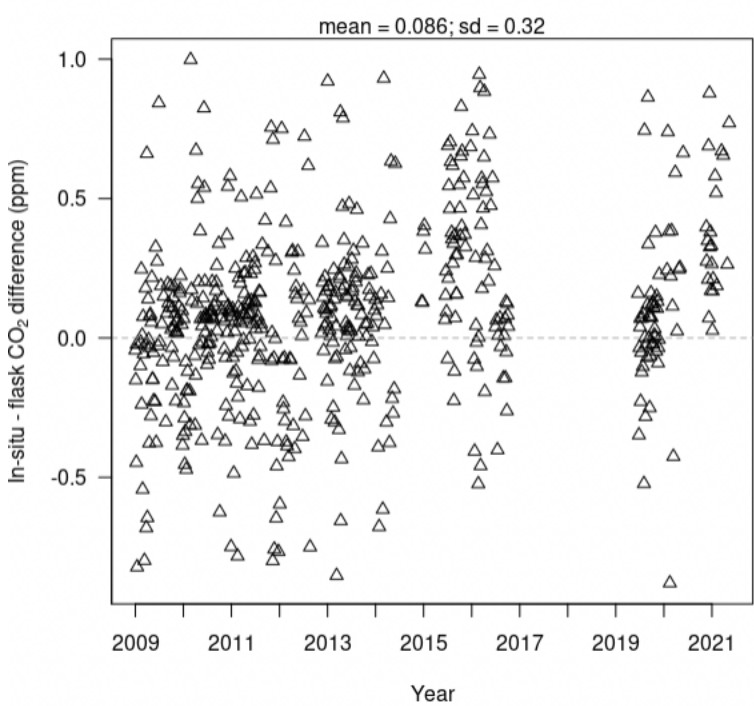

Figure 2. Differences between deconvolved (Section 2.1) in-situ and flask data for CO₂.

**Figure 2. Differences between deconvolved (Section 2.1) in-situ and flask data for $CO_2$.**

## 2.2 Signal Processing

This study utilised only the daytime measurements averaged daily over the period from 13:00 to 17:00 local time and measured at the height of 301 m. The use of such daytime-only values from the top of the ZOTTO tower ensures that the measurements are representative of the air in the entire boundary layer, thereby avoiding the domination of full daily averages by the high $CO_2$ mole fraction during night-time due to the accumulation of $CO_2$ in the shallow stable boundary layer as a result of the night-time temperature inversion.

The variability in $CO_2$ measurements at specific locations around the globe can, in general, be described as a combination of three signals: a long-term trend, a non-sinusoidal yearly cycle reflecting the change of the seasons, and short-term variations associated with meteorological variability that can last from several hours to several weeks as a result of local and regional influences. In this study, we use the CCGCRV (Thoning et al., 1989) curve-fitting application to separate these signals in the ZOTTO observations. CCGCRV (Thoning et al., 1989) is a curve-fitting application for long-lived GHG time series maintained at the Carbon Cycle Group of Climate Monitoring and Diagnostics Laboratory (CCG/CMDL) of the National Oceanic and Atmospheric Administration (NOAA, USA). The version of CCGCRV used here is applied as a stand-alone

function in Python and is available from the NOAA CMDL ftp server at: https://gml.noaa.gov/aftp/user/thoning/ccgcrv/. The result of the curve fitting method is a function fit to the data, which approximates the annual oscillation and the long-term growth in the data, represented by a polynomial function and harmonics of a yearly cycle, respectively. The fit function includes digital filtering of the residuals from the fit by short-term and long-term cut-offs in units of time (i.e., days) to define interannual and short-term variations that are not determined by the function.

Since the curve-fitting method is sensitive to its parameter settings (Pickers and Manning, 2015), we created an ensemble of 100 curve-fitting settings with the three polynomials, with the number of harmonics varying from 2 to 6. For each harmonic option, 20 short-term and long-term cut-off values were randomly chosen from 88 days to 240 days and from 667 days to 800 days, respectively (Table B1). To remove the impact of unreliable $CO_2$ observations, which can strongly affect estimates of the seasonal cycle, for each ensemble member, any data lying outside the range defined by three times the normalised root-mean-square deviation relative to the smooth curve were iteratively discarded from the original time series until all outliers were removed (Kozlova et al., 2008). The percentage of the removal data is 2.4% of the total data.

## 2.3 Estimation of the duration and amplitude of the Carbon Uptake Period (CUP) and carbon release period (CRP)

In this study, we apply a method to estimate CUP and CRP presented in Kariyathan et al. (2023). We calculate the first derivative of the $CO_2$ mole fraction with respect to time at every (daily) data point. To determine the CUP (CRP) onset and termination, we determined the point in time at which the time series of the first derivative crosses a threshold defined as a given percentage of the annual minimum (maximum) of the first derivative (Fig. 3). The difference in time between the defined onset and termination then represents the length of CUP (CRP). The absolute difference in mole fraction between these two points represents the amplitude of CUP (CRP). The rate of uptake (release) is then simply calculated by dividing amplitude by the length of CUP (CRP), assuming that the change in the curve shape of the season over the years is negligible. The use of the time derivative of a time series can provide a more robust estimate of the key dates that define the CUP and CRP without taking into account the changes in the shape of the seasonal curve compared to the conventional use of zero crossing date derived from the detrended $CO_2$ seasonal cycle (Barlow et al., 2015) (More in Section 3.1).

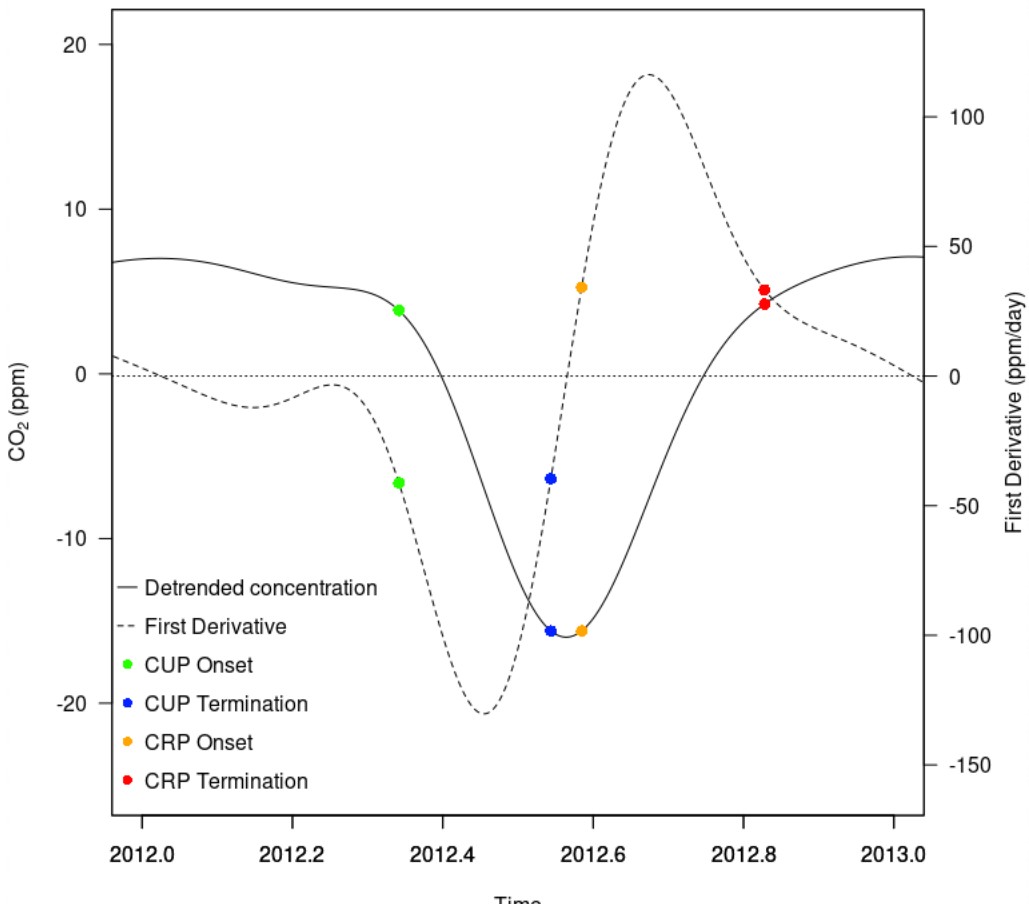

**Figure 3. A complete CO$_2$ seasonal cycle for 2012 (solid line) and its first derivative representing the rate of CO$_2$ uptake or release (dashed line). The blue and green points mark the time of 30% of the minimum of the first derivative, while the red and orange points are 30% of the maximum of the first derivative.**

For the analysis shown here, we use a threshold of 30% of the first derivative minimum. ZOTTO is a high-latitude, continental station therefore the seasonal cycle of CO$_2$ is characterised by a flat maximum, showing one or even multiple peaks during the winter (Piao et al., 2008). This procedure results in several zero-crossings in the first derivative during the winter-to-spring seasonal transition. After conducting several tests with various threshold values, we decided that for the ZOTTO dataset, the value needed to be at least higher than 15% to avoid multiple zero-crossings in the first derivative and thus clearly identify the timings of CUP. To assess the impact of the chosen threshold value on our results, we also varied the threshold between 15%-30%. The different thresholds influence the absolute length and amplitude of CUP without affecting the anomalies across years or the long-term trend (results not shown).

## 2.4 Jena CarboScope Global Inversion Set-up

We derived spatio-temporal variations of Net Ecosystem Exchange (NEE) from long-term atmospheric $CO_2$ measurements using a Bayesian atmospheric $CO_2$ inversion of atmospheric transport (Jena CarboScope, Rödenbeck et al. 2003, updated). The inversion procedure uses the atmospheric tracer transport simulated by the TM3 model (Heimann and Könner, 2003) (resolution = 4 degrees x 5 degrees x 29 layers) driven by meteorological fields from the NCEP reanalysis (Kalnay et al., 1996). Using the atmospheric tracer transport model to simulate the atmospheric $CO_2$ field that would arise from a given flux

field, the inversion algorithm finds the flux field that leads to the closest match between observed and simulated $CO_2$ mole fractions. In addition, the estimation is regularised by a priori constraints meant to suppress excessive spatial and high-frequency variability in the flux field. The a priori settings do not involve any information from biosphere process models. Fossil fuel fluxes are fixed to accounting-based values (Jones et al., 2021). Ocean fluxes are fixed to the CarboScope estimates oc_v2022 (Rödenbeck et al., 2013, updated) based on an interpolation of $pCO_2$ data from SOCATv2022 (Bakker et al., 2023).

We performed three inversion runs, listed in Table 1, spanning 2005-2021, i.e., including the 2010-2021 study period preceded by five years spin-up time to account for any initialisation processes of the model. The three inversions differ in the set of atmospheric measurement stations used. The run labelled s10v2022 uses 78 globally distributed atmospheric monitoring stations, which does not include any stations in Siberia (Fig. D1 and Table D1). To assess the impact of the contribution of the ZOTTO data on the estimated flux inferred by the inversion model, we additionally performed an inversion (s10v2022+ZOT)

in which we added the continuous atmospheric $CO_2$ observations at 301 m from the ZOTTO station to the station set. The third inversion (s10v2022+Allstations) also includes further atmospheric monitoring stations in Siberia, namely Tiksi (TIK at 71°60′ N, 128°89′ E ranging 2011-2019), Noyabrsk (NOY at 63°43′ N, 75°78 E′ ranging 2005-2019), Demyanskoe (DEM 59°79′ N, 70°87′ E ranging 2005-2019), Karasevoe (KRS 58°25′ N, 82°42′ E ranging 2004-2019), Azovo (AZV 54°71′ N, 73°03′ E ranging 2007-2019) (Fig. D1 and Table D1). From the s10v2022+Allstations inversion, we only analyse the 2011-2019 period

since this is the time when all involved atmospheric sites actually have data.

**Table 1. Inversion runs used in this study.**

| Label in figures | Calculation Period | Atm. Sites |
|---|---|---|
| **s10v2022** | 2005-2021 | 78 (s10v2022) |
| **s10v2022+ZOT** | 2005-2021 | 79 (s10v2022+ZOT) |
| **s10v2022+Allstations** | 2005-2019 | 84 (s10v2022+ZOT+TIK+KRS+NOY+DEM+AZV) |

**2.5 Flux area and derivation of CUP and CRP from posterior NEE fluxes**

To understand to what extent the interannual variations in CUP and CRP observed in the ZOTTO data are explainable by regional ecosystem responses to interannual climate anomalies, we first determine the "region of influence" of the ZOTTO data on the NEE derived from the CarboScope inversion. We approximate this by calculating the 2010-2021 spatial root mean square (RMS) of the climatology monthly difference between the inverted NEE without (s10v2022) and with (s10v2022+ZOT) ZOTTO data included. The approximate region of influence of the ZOTTO data on estimates of net ecosystem exchange is then determined by 40% of the average of all the monthly RMS differences (The red shading in Fig. D1). The ecosystem cover in this region of influence comprises Pinus sylvestris forest stands (about 20 m in height) on lichen-covered sandy soils. We then aggregated the NEE fluxes derived from the inversion with ZOTTO data included (s10v2022+ZOT) for this region. The first derivative of the atmospheric $CO_2$ data corresponds to the net land and ocean $CO_2$ flux, and assuming insignificant imprint of variations in fossil and ocean fluxes, to the NEE derived from the atmospheric inversion (see also discussion). Therefore, similar to the process applied for the observed dataset, to determine the timings and length of CUP (and CRP) from posterior NEE flux, all the data points before and after the flux minimum (maximum) when the flux value is less than 30% of the minimum (maximum) NEE were selected. The amplitudes of CUP and CRP are the integrals of the fluxes between the onset and termination of CUP and CRP.

**2.6 Partial Correlations with Climate Anomalies**

We calculated partial correlation coefficients between seasonal temperature anomalies and the timing and intensities (i.e., length and amplitude) of CUP (and CRP). To quantify the decadal change in the partial correlation, we controlled for the effects of precipitation and cloud cover. We used monthly climatic data (temperature (in ° Celsius), precipitation (in mm day⁻¹), and cloud cover (in percent)) at a spatial resolution of 0.5° for 2010-2021 from ERA5 reanalysis (Hersbach et al., 2020). The region of influence used for this correlation analysis (red shading area in Fig. D1) was derived from Section 2.5 with additional spatial weighting by the annual Gross Primary Production (GPP) from the observation-derived upscaling product by Jung (2011) to focus the integration of the climate data on the vegetated areas and remain independent from the potential biases of the inversion.

**3 Results and Discussions:**

The ZOTTO daytime data at 301 m used in this study are presented in Fig. 4a. The 2010-2021 average amplitude of the seasonal cycle at ZOTTO calculated from CCGCRV smoothed mole fraction is 25.5 ppm, after subtracting the linear part from the harmonic fitting. This number is comparable with previously reported values of 26.6 ppm at ZOTTO in the year 2007 (Kozlova et al., 2008). Due to its continental location, the amplitude at ZOTTO is larger than at other tall tower sites with stronger marine influence, e.g., Bialystok, Poland, with 23 ppm (Popa, 2007), or Ochsenkopf, Germany, with 15.5 ppm

(Thompson et al., 2009) at the uppermost tower levels (300 and 163 m a.g.l., respectively). The seasonal amplitude of the ZOTTO mole fraction data is also more prominent than that of the Marine Boundary Layer product (MBL) (NOAA 2022),

which is based on measurements from National Oceanic and Atmospheric Administration (NOAA) Cooperative Global Air Sampling Network sites, where samples are predominantly of well-mixed marine boundary layer but at the same latitude as ZOTTO and away from anthropogenic and strong natural sources and sinks (Fig. 4b). A clear pattern becomes evident: during winter, $CO_2$ is released over continents by biospheric respiration and anthropogenic emissions, while the ocean may even counteract the global $CO_2$ increase through increased $CO_2$ solubility in cold water. Fig. 4 also shows the $CO_2$ uptake during

summer: The photosynthetic uptake from the biosphere over the continent amplifies the summer minimum in the $CO_2$ data and produces an offset relative to the MBL data. Moreover, a time shift in the summer minima can be seen between ZOTTO and the maritime background. This time shift is induced by the transport of the $CO_2$-depleted air from the continents towards the ocean. Annual $CO_2$ mole fractions at ZOTTO are higher and the interannual $CO_2$ growth rate exhibits a stronger variability than the MBL. This indicates the strong continental influences of ZOTTO location in Central Siberia. Therefore, it is important

to investigate the trends of different components of the annual seasonal cycles as well as the implications for the continental carbon cycle.

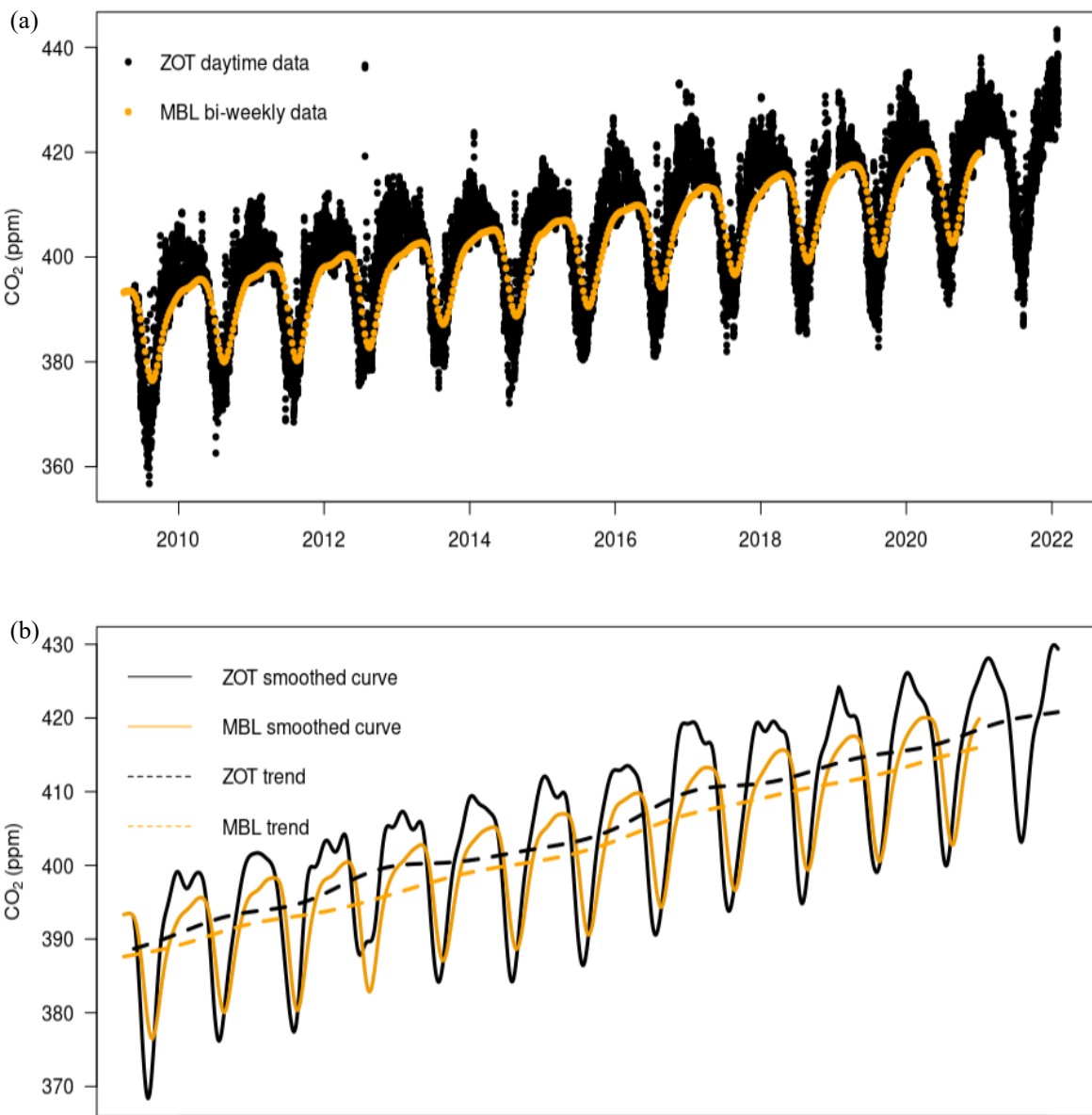

**Figure 4. (a) Daytime CO$_2$ time series of the continental ZOTTO station in comparison to the biweekly marine boundary layer (MBL) (b) Time series of (a) smoothed by the Thoning et al. (1989) algorithm.**

**3.1 Interannual Variation in the Timing and the Intensity of the Carbon Release Period (CRP) and Carbon Uptake Period (CUP).**

The analysis of data for the 11 years of observations at ZOTTO reveals that the annual minimum of $CO_2$ in the individual years of the smoothed time series was registered on one of the days during the period from July 26 to August 3, whereas the annual maximum was registered within a wider time period: from December 23 to January 29 (Fig. 4). The mean onset and termination dates for the CUP over 2010-2021 are May 01 and July 20 respectively. Similarly, the mean onset and termination for the CRP are on August 02 and January 02.

There were no significant trends in the timing of CUP (i.e., onset and termination) (Fig. 5). This finding is opposite to the earlier studies by Piao et al. (2008) and Barichivich et al. (2012), showing results from the 20-year atmospheric $CO_2$ mole fraction data record from high-latitude stations in Alaska and Canada and the CarbonTracker data assimilation system. These studies found a trend towards an earlier onset of autumn-to-winter carbon dioxide build-up for the period 1990-2010, suggesting a shorter net carbon uptake period. Our finding is similar to that of Liu et al. (2018), in which they show a reduction in the response of decomposition to warming for the 1997-2011 period, suggesting that autumn warming in boreal and arctic ecosystems no longer advances the end of the carbon uptake period as previously suggested. Notwithstanding, it is important to take into account the shorter time span of the ZOTTO dataset when making comparisons to results from other studies with longer measurement records. The absence of a significant trend in the timings of CUP (i.e., onset and termination) occurs in the light of significant interannual variability in the timing of these events. For instance, there was an abnormally early onset of CUP in 2020 (Fig. 5). This finding will be further analysed later in this section. Figure 5 also shows a significant increasing trend in the timing of release termination, suggesting the termination of CRP was happening later and later during the 11-year study period.

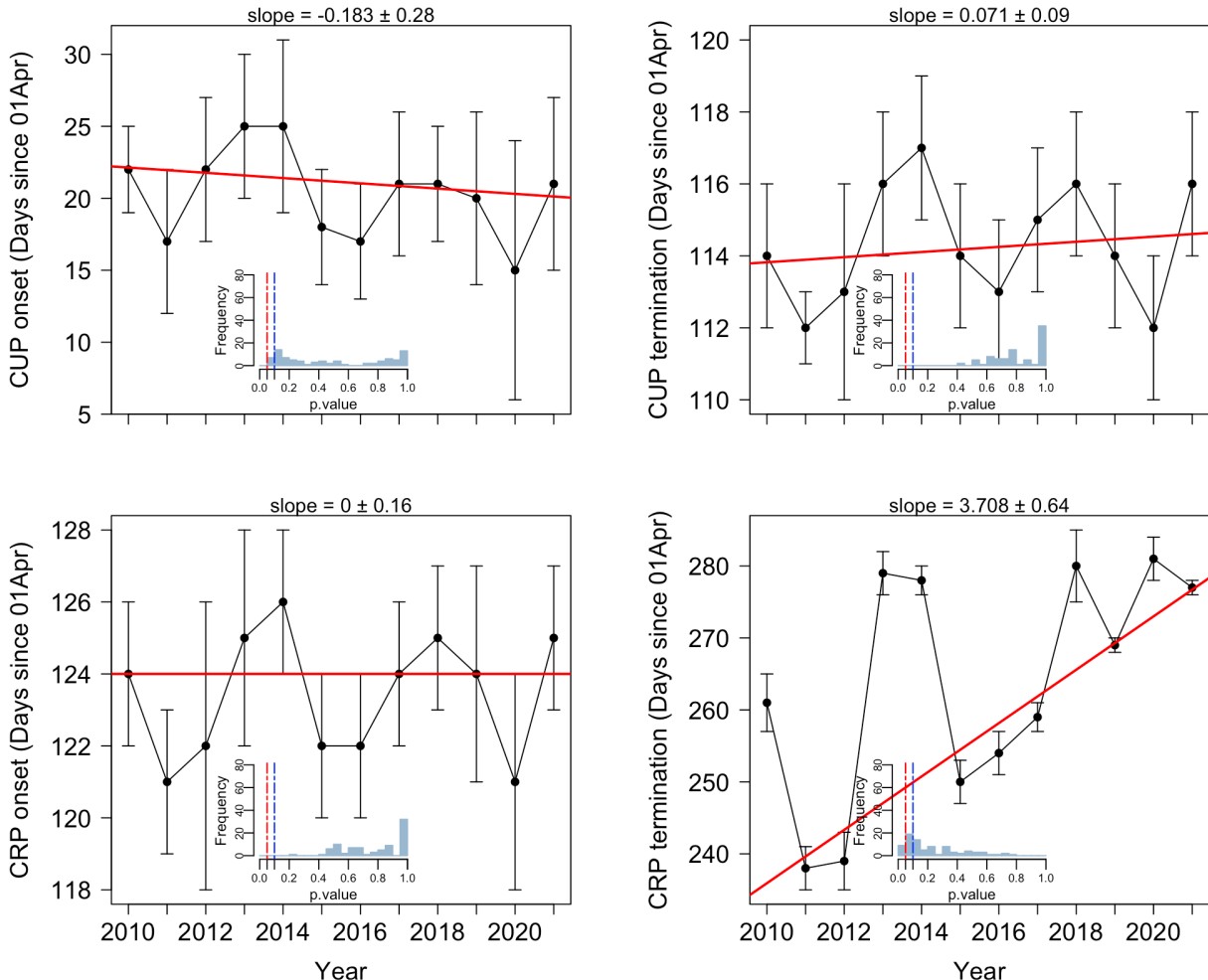

**Figure 5. Time series of the timings of CRP and CUP's onset and termination. Inset histograms show the frequency of p-values of the Theil-Sen regressions of the 100-member curve-fitting ensemble, with red and blue vertical dashed lines, respectively, indicating the 0.05 and 0.1 significant difference from 0 of the slopes. Error bars are the standard deviations from the mean of the 100-member curve-fitting ensemble.**

We found a significant negative correlation between spring (MAM) temperature anomalies and the onset of CUP (Fig. 6) (R = -0.52, p < 0.05). Similarly, summer temperature (JJA) anomalies were negatively correlated with the onset of the CRP (Fig. 6) (R = -0.48, p < 0.05). The termination of CRP is positively correlated with the proceeding autumn (SON) temperature anomaly (R = 0.53, p < 0.1) (Fig. 6).

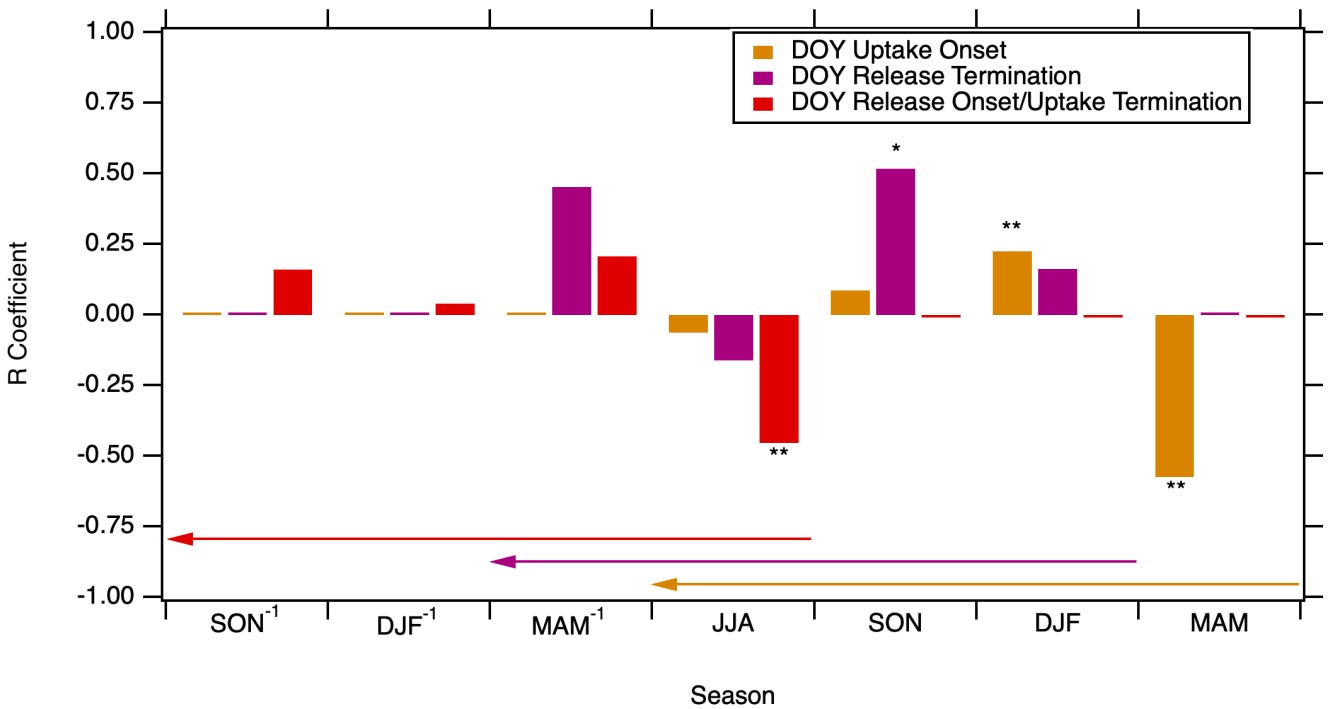

**Figure 6. The partial correlation coefficient between seasonal temperature anomalies and the timing of release start (or uptake termination), release termination and uptake onset controlling for the effects of precipitation and cloud cover over the 2010-2021 period. The timings are correlated with the seasons they fall in and also with three preceding seasons (demonstrated as the length and direction of the arrow of the respective colour). We calculate the partial correlations by selecting every subset of 10 years in the 11-year 2010-2021 period (omitting one year in each calculation) and taking their standard deviation as the error bar. Errors bars are small and therefore not visible in this figure. Bars marked * and ** indicate that the partial-correlation coefficient is significant at p < 0.1 and p < 0.05, respectively.**

There were clear, significant increasing trends in the release length and release amplitude (at $p < 0.1$ and $p < 0.05$ level, respectively) (Fig. 7). For the CUP, both the uptake length and amplitude also increased significantly (at $p < 0.05$ level and 0.1 level, respectively) over the study period. The trend in the amplitude was interrupted by two years (2012 and 2020) with anomalously small amplitude. Taken together, these trends provide evidence for the amplification of the seasonality of atmospheric $CO_2$ at ZOTTO. There were abnormal decreases in both CUP and CRP amplitude in 2012 (Fig. 7). Without the abnormal years 2012 and 2020, the trends of CUP and CRP amplitude would be 2.43 ppm year$^{-1}$ and 1.93 ppm year$^{-1}$, respectively. Our finding is consistent with Graven et al. (2013), comparing 2009-2011 aircraft-based observations of $CO_2$ above the North Pacific and Arctic Oceans to earlier data from 1958 to 1961 and found that the seasonal amplitude at altitudes of 3 to 6 km increased by 50% for high latitudes. Forkel et al. (2016) combined observations and models showing that climate warming has caused an increase in carbon uptake amplitude as a result of the global $CO_2$ fertilisation effect. This led the uptake

rate of carbon to increase faster than its respiratory release rate from the terrestrial biosphere. However, in this study, the slopes or the rates of uptake and release stay the same despite both the amplitude and length of CUP and CRP increasing significantly over the 11-year studying period. This could suggest that the increase in either the amplitude or the length of CUP and CRP observed at ZOTTO is not strong or dominant enough to alter the rate of uptake and release over the 11-year period.

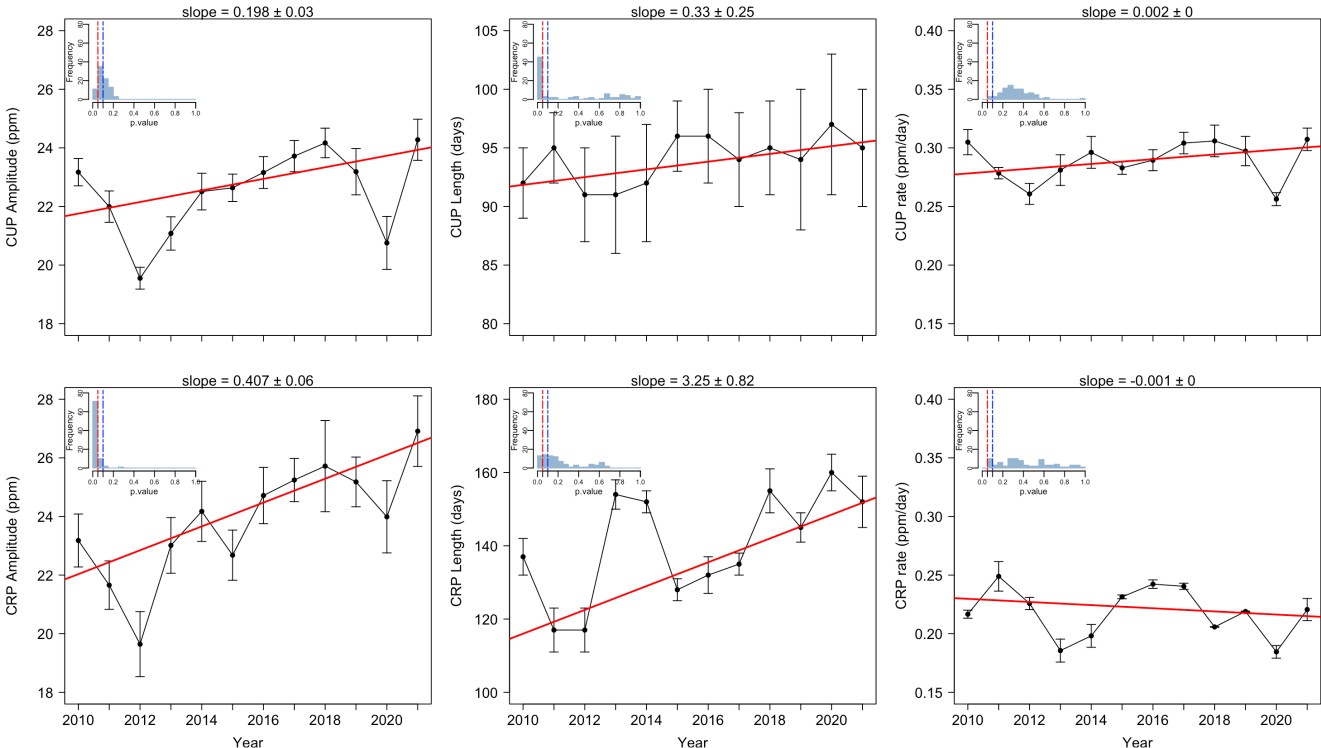

**Figure 7. Time series of the length and amplitude of CRP and CUP. Inset histograms show the frequency of p-values of the Theil-Sen regressions of the 100-member curve-fitting ensemble, with red and blue vertical dashed lines, respectively, indicating the 0.05 and 0.1 significant difference from 0 of the slope. Error bars are the standard deviations from the mean of the 100-member curve-fitting ensemble.**

Positive correlations between spring temperature and CUP's amplitude and length ($R = 0.56$, $p < 0.05$ and $R = 0.7$, $p < 0.05$ respectively) were stronger than those between autumn and winter temperatures and CRP's amplitude and length ($R = 0.52$, $p < 0.05$, $R = 0.19$, $p < 0.05$ respectively) (Fig. 8). Summer (JJA) temperature anomaly was also significantly negatively correlated with CUP amplitude (Fig. 8).

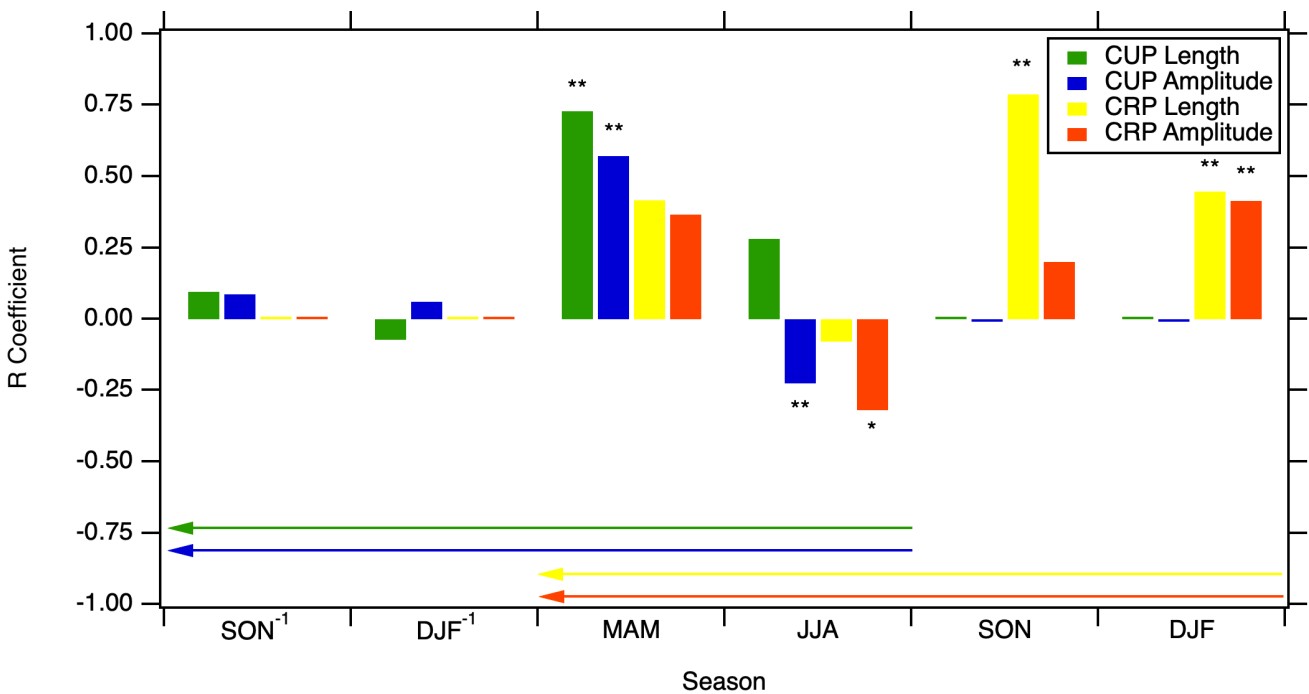

**Figure 8. The partial correlation coefficients between seasonal temperature anomalies and the amplitude and length of CUP and CRP, controlling for the effects of precipitation and cloud cover over the 2010-2021 period. The amplitude and length are correlated with the season when CRP and CUP end and also with three preceding seasons (demonstrated as the length and direction of the arrow of the respective colour). We calculate the partial correlations by selecting every subset of 10 years in the 11-year 2010-2021 period (omitting one year in each calculation) and taking their standard deviation as the error bar. Errors bars are small and therefore not visible in this figure. \* and \*\* indicate that the partial-correlation coefficient is significant at P < 0.1 and P < 0.05, respectively.**

Our correlation analyses suggest that warmer temperatures during the spring growing season are linked to earlier spring phenology in temperate and boreal forests (Fig. 6), as also shown in Gu et al. (2022). These results are also consistent with Barichivich et al. (2012), who used gridded daily temperature from 1950 to 2011 and atmospheric $CO_2$ mole fraction data from high-latitude observing stations and the CarbonTracker assimilation system, showing that higher late summer/early autumn temperatures lead to an earlier onset of autumn carbon release while higher growing-season temperatures lead to an earlier onset of spring carbon uptake. This finding coincides with an unprecedentedly strong and persistent heatwave in the winter to spring of 2020 in Siberia that resulted in an early onset of CUP, as seen in Fig. 5. A warmer spring could potentially increase the carbon uptake amplitude, as seen in Fig. 8. However, during 2020 when the Siberian winter-to-spring heatwave occurred there was only an increase in the CUP length due to the early spring onset but not in the CUP amplitude (Fig. 7). Our finding matches the results from the recent study of Kwon et al. (2021) where they found that during the Siberian 2020 heat wave, the

warming-induced enhanced photosynthetic $CO_2$ uptake in spring was offset by a larger reduction in $CO_2$ uptake in late summer due to soil moisture deficit, resulting in the mean annual $CO_2$ uptake over Siberia slightly lower than the average of the previous five years. The warmer 2020 spring conditions promoted increased vegetation growth, which, in turn, contributed to fast soil moisture depletion, causing plants to close their stomata to conserve water. This led to reductions in evapotranspiration and photosynthetic activity, thereby reducing carbon uptake from the atmosphere in the later summer. Additionally, with an early onset of the growing season and the warm temperatures during summer, the active soil layer will tend to get deeper in late summer and fall (Fisher et al., 2016). This will, in turn, lower the water table and, in some cases, remove the soil water pool far enough from the rooting zone to cause a draught effect (Costa et al., 2023).

The Siberian wildfire in the late summer of 2012 resulted from dryness. Low moisture pre- and post-fire could have led to reductions seen in both CUP and CRP amplitude in 2012. Indeed, for much of 2012, there was an abnormally high summertime CO observed at ZOTTO (Fig. E1), which would confirm a very strong fire season. There are clear increasing trends in the CUP and CRP amplitude, but it cannot be ruled out that extremes and the legacy effect of ecosystem recovery from the 2012 wildfires impact these trends to a significant degree (i.e., the increasing trend that we observed might not have been as strong and significant as without the legacy effect). In the same analysis derived from HPspline curve fitting (Fig. C1 and C2), the CUP amplitude trend does not have such as strong "recovery" after 2013. The legacy effect, thus, perhaps only occurred in 2013.

For our study, we applied an alternative method that derived seasonal components from the first derivative of the mole fraction data, as in Kariyathan et al. (2023) (described in more detail in Section 2.3). This method was shown to give a more robust estimation of CUP duration than the conventional "zero-crossing method" (Barlow et al., 2015). Previous studies have used the zero-crossing times (i.e., the downward and upward $CO_2$ zero-crossing dates as the day on which the detrended curve crossed the zero line from positive to negative and from negative to positive, respectively) and their difference as proxies for the onset, termination, and duration of the net CUP. This approximation assumes that the shape of the seasonal cycle does not change significantly, and hence, a change in the phase at one point (e.g., maximum) of the seasonal cycle provides information on phase changes at other points (Barichivich et al., 2012). However, the zero-crossing times may not be the best proxy if the shape of the seasonal cycle changes substantially from year to year or when the seasonal cycle is not symmetric around the maximum/minimum (skewed seasonal cycle) (Barlow et al., 2015; Kariyathan et al., 2023).

The limitation of the CCGCRV or other existing harmonic-based curve-fitting methods is their limited ability to properly address non-stationary processes, which is most noticeable during rapidly changing environmental conditions (e.g. drought, heatwave) that affect the amplitude and phase of the seasonal time series. However, considering this limitation, the CCGCRV-derived smoothed time series for ZOTTO data still represented anomalous seasons (Fig 4b.). To ensure that results are not unduly influenced by the mathematics underlying this specific curve fitting program, we repeated our analysis using detrended

data derived from an alternative curve fitting program – HPspline (Keeling et al., 1986) (Fig. C1 and C2). There were no significant differences in seasonal signals between the two curve-fitting programs.

### 3.2 Analysis from Jena CarboScope global inversion.

Before analysing posterior NEE fluxes, we compared the trend and interannual variability of the timing and intensity of CUP and CRP derived directly from observed data with the atmospheric mole fraction simulated by TM3 based on the CarboScope posterior NEE fluxes from the inversion (s10v2021+ZOT) to ascertain that the inversion captures the observed patterns discussed in Section 3.1 (More in Appendix H). In general, the inversion is capable of well reproducing the inter-annual variations and trends resulting from the combination of variability in atmospheric transport (derived from meteorological variations) and regional ecosystem flux responses to climate variations (Fig. H1-H6). This gives us more confidence in the posterior NEE fluxes derived from the model that we will now use to further assess the signals and variations we have seen in the observation analyses.

In section 3.1, we have shown that the variations in CUP and CRP timings and intensity derived from mole fraction data correlate with climate anomalies in the region. The flux anomalies inferred from the inversion using the ZOTTO data averaged for the "region of influence" broadly lack similarity in interannual variability between the timing of CUP and CRP derived from mole fraction data, with the notable exception of the CUP termination, while the trends over the 11-year period are not statistically significant in both inversion-based NEE and mole fraction data (Fig. 9). One exception is that the posterior NEE shows early onset of CUP in 2020, even though the magnitude of this early onset is not as large as seen in the mole fraction analysis derived from ZOTTO measurements.

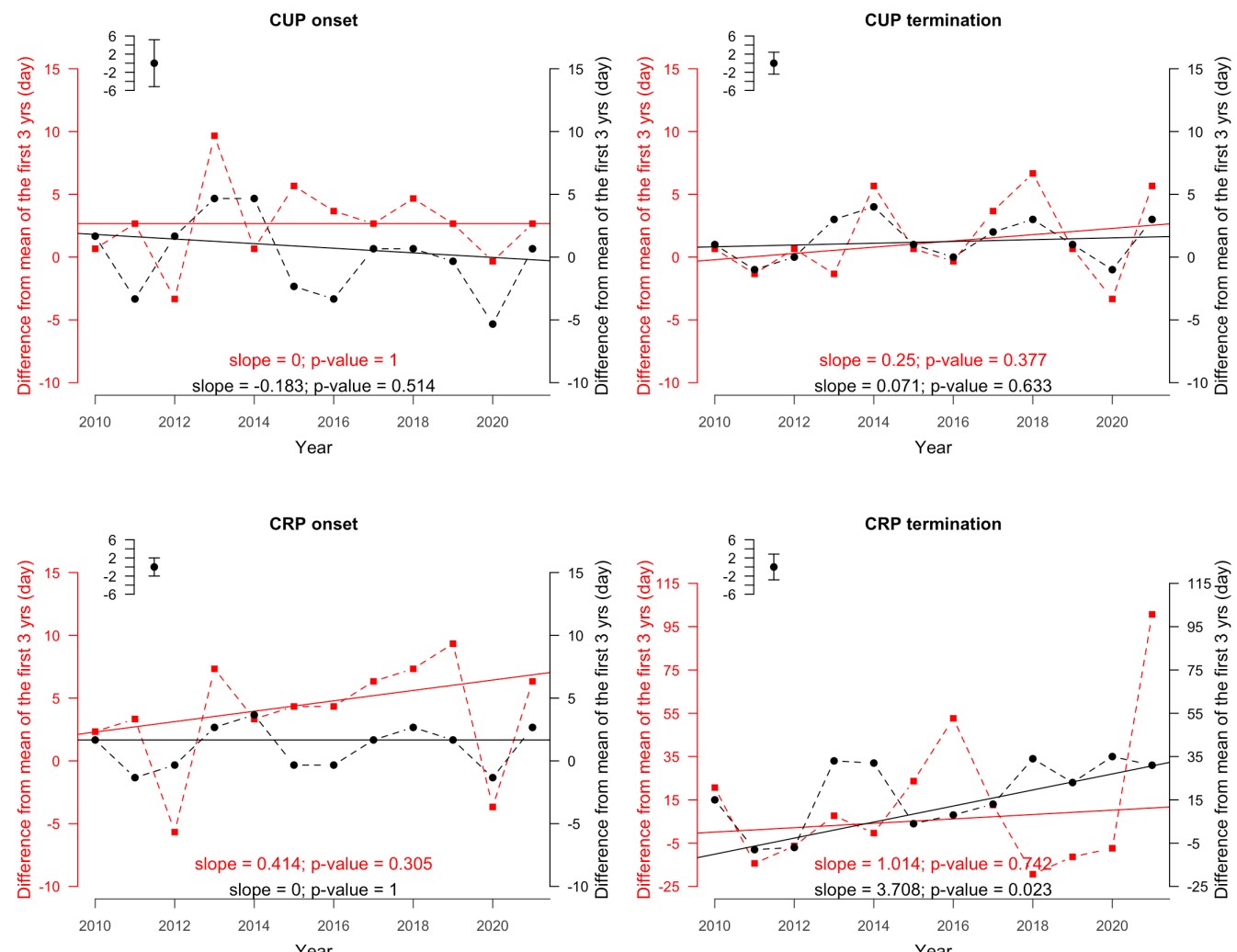

**Figure 9. Time series of the timing of CRP and CUP derived from observational atmospheric mole fractions (black circle) and regional fluxes (red square). The small bar plot on the top left of each panel indicates the 2010-2021 average error of the observational analysis. We calculated the difference from the mean of the first three years (in days) to take into account the offset in scale between the two analyses, therefore, better visualise the inter-annual variations of each analysis.**

The CUP and CRP lengths derived from NEE fluxes are shorter compared to those derived from the atmospheric $CO_2$ mole fractions, resulting from the difference in variabilities in the timing in onset and termination of CUP and CRP between the two analyses (Fig. 10). The anomalies 2012 and 2020 shown in the uptake and release amplitudes as inferred from the observed mole fractions are not apparent when analysing the regional NEE fluxes. This difference occurs despite the fact that the 2012

and 2020 anomalies do exist in the atmospheric $CO_2$ mole fraction simulated from the fluxes estimated by the inversion (Fig. H2). One possible interpretation of this finding is that the regional NEE fluxes were not significantly influenced by the wildfire and heat wave in 2012 and 2020, respectively. This suggests that for the inversion, there is not sufficient constraint on interannual regional flux variations to attribute the strong mole fraction anomalies in 2012 and 2020 to regional signals – instead, the inversion suggests that these are likely the consequence of a hemispheric instead of a central Siberian signal. We cannot exclude the possibility that the area of influence of the ZOTTO data, defined in this paper based on the 11-year average imprint of ZOTTO in the inversion, is not well representing the influence areas under these anomalous climate conditions, leading to an erroneous attribution of the flux anomaly to hemispheric scales. To further investigate and confirm this, a footprint analysis from a transport model or a regional inversion study during these abnormal occasions or the use of a multiple-factor inversion (such as the NBE-T inversion by Rödenbeck et al. (2018)) is needed, which is outside the scope of this paper.

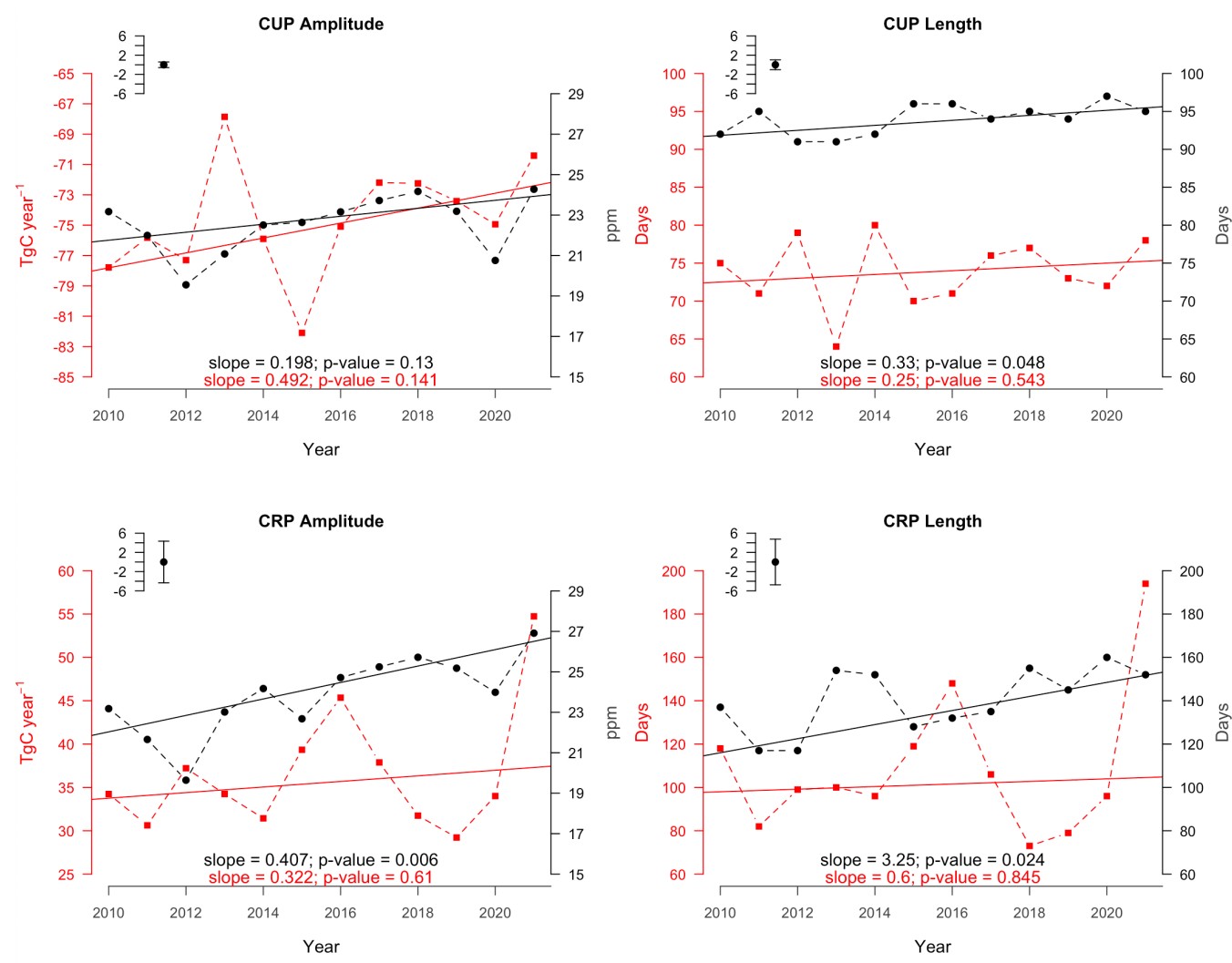

**Figure 10. Time series of the amplitudes and lengths of CRP and CUP derived from atmospheric mole fractions (black circle) and regional fluxes (red square). The small bar plot on the top left of each panel indicates the 2010-2021 averaged error of the observational analysis.**

We also compared our main analysis (i.e., timing and intensity of CUP and CRP) among all three inversions listed in Table 1. In terms of the timing analysis, apart from CRP termination, the trends among all runs are quite similar (± 0.2) (Fig. G1). The year-to-year variations seem to become more prominent as more stations are added into the inversion. CRP termination derived from the run with added ZOTTO data has an abnormal value in 2021. With regards to the intensity analysis (Fig. G2), the uptake and release amplitudes decrease (in absolute terms) as more stations are added into the model, and the length of these periods also decreases.

To understand more about the inferred NEE fluxes that we used in the above analyses, we will now compare the two inversions using station sets s10v2021 and s10v2021+ZOT. The assessment of the impact of the ZOTTO station on the inverted NEE (based on the comparison between our different Jena CarboScope inversions with and without ZOTTO) can help to separate the contribution of the influence of meteorological variability and regional ecosystem NEE on the observed mole fraction.

The 2010-2021 averaged annual Northern Hemisphere (NH) (>30 °N) NEE values for the inversions using station sets s10v2021 and s10v2021+ZOT are of comparable magnitude with -0.32 and -0.31 PgC year$^{-1}$ respectively. The use of ZOTTO data in the inversion reduces the seasonal amplitude of the NEE within the area of influence of ZOTTO as defined in the Methods (Fig. 11). The 11-year (2010-2021) averaged annual NEE values of the constrained region (Fig. D1) for the inversions using station sets s10v2021 and s10v2021+ZOT are -50 and -30 TgC year$^{-1}$, respectively. The percentage difference in the 2010-2021 cumulative NEE flux between estimates with and without the ZOTTO station dataset into the global inversion (i.e., s10v2022 vs s10v2022+ZOT) is ~39% (Fig. G3), i.e., weaker regional uptake when using ZOTTO data. Since the global carbon budget is closed at interannual timescales, adding ZOTTO data to the inversion altered the estimated carbon uptake within the rest of the world accordingly to conserve mass, leading to higher carbon uptake spread widely across the NH tropical and mid-latitude 20°-50°N (Fig. G4). To compare the interannual variability of NEE within the defined region of influence, we also calculate the average NEE for each year and then calculate the standard deviations (SD) of these values. The SD values of the inversions using station sets s10v2021 and s10v2021+ZOT are 6 and 9 Tg C year$^{-1}$ respectively. Adding ZOTTO data increases the variability attributed to the region. The ZOTTO data corrects for biases in the standard inversion and leads to a more correct representation of amplitude (i.e., gross photosynthesis – respiration) and the imbalance decadal NEE, therefore better optimising the flux in the constrained region.

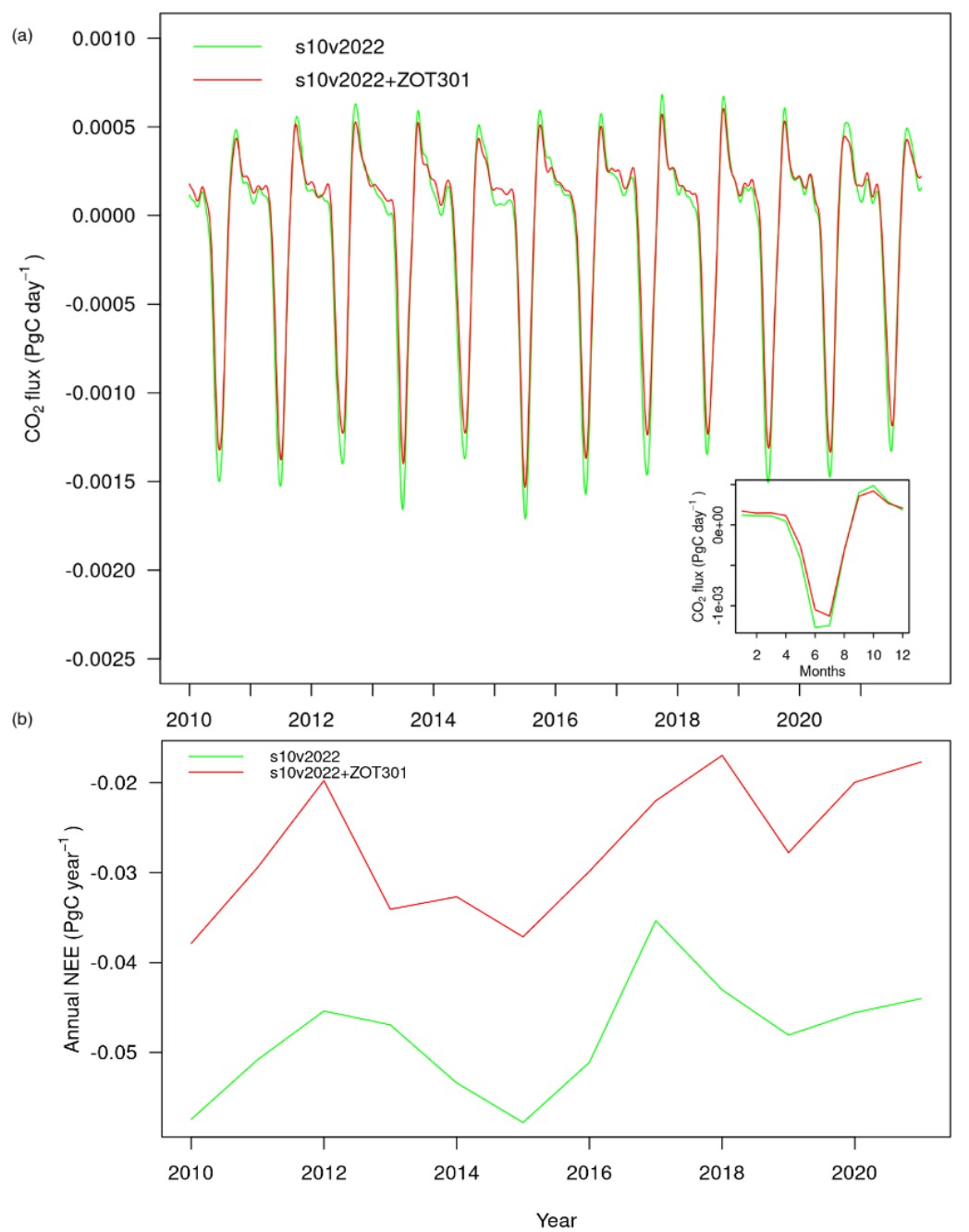

**Figure 11. NEE estimated (a) daily and (b) annually by the inversions using the station sets s10v2022 (red) and s10v2022+ZOT301 (green). The small panel on the bottom right corner of (a) is 2010-2022 averaged monthly NEE from the two inversions.**

## 4 Conclusions

Our analysis of the continuous record of $CO_2$ mole fractions at ZOTTO shows the high quality of the data with no systematic error over the period 2009-2022. The data reveals that the $CO_2$ uptake and release amplitude and length significantly increased from 2010 to 2021, where the increasing trend in CRP amplitude is bigger than that of the CUP. This pattern corresponds well to the global trend of increased intensity of the seasonality in northern hemispheric carbon exchange. The data shows a strong negative correlation between spring temperature and CUP onset, as well as between late summer temperature and CUP termination/CRP onset, suggesting a strong regional influence of local climate on the observed mole fractions. However, there were no significant trends in the timing of $CO_2$ uptake and release in our 11-year study period.

We have shown through mole fraction analyses the influences of two extreme events, the wildfires in 2012 and the 2020 heat wave. However, the inversion-based NEE fluxes using the ZOTTO data did not show the flux anomalies deriving from the Siberian wildfires in 2012 and the 2020 Siberian heat wave as seen in the observational analyses: the interannual variations from the NEE flux analysis were different from that from the mole fraction analyses. This could suggest that the variabilities that are only seen in the atmospheric mole fraction analyses could be derived from outside the defined area of influence of ZOTTO. However, we cannot rule out the possibility that the weight of ZOTTO data in the inversion compared to other regions of the world and the absence of sufficiently long, continuous $CO_2$ mole fraction measurements in other Siberian regions prevent a robust attribution of Siberian variability by the inversion. Possibly, the quantification of regional fluxes could be improved by using satellite data collected during summer months when the observing geometry is favourable but also during other months via an improved higher-resolution regional transport model or the use of additional constraints in the inversion such as climate anomalies (Rödenbeck et al., 2018). Due to the sparseness and uneven distribution of the monitoring surface networks, it is still debatable whether a higher resolution regional transport model alone may better constrain regional fluxes.

**Appendices**

**Appendix A: Calibration tanks system.**

**Table A1. Calibration tank system at ZOTTO.**

| Tank name | ID number | $CO_2$ (ppm) | $CH_4$ (ppb) |
|---|---|---|---|
| Calibration Tank 1 | D478665 | $354.87 \pm 0.06$ | $1804.0 \pm 1.6$ |
| Calibration Tank 2 | D436606 | $394.81 \pm 0.06$ | $1898.1 \pm 1.4$ |
| Calibration Tank 3 | D436607 | $453.56 \pm 0.09$ | $2294.1 \pm 2.2$ |
| Target Tank | D478666 | $404.60 \pm 0.09$ | $1946.4 \pm 1.5$ |

**Appendix B: CCGCRV parameter settings**

**Table B1**. **Ranges of input parameter settings of the CCGCRV smoothing algorithm that were used to test program sensitivity.**

| Parameter | Range of values tested |
|---|---|
| Short-term cut-off period, in days | 88, 95, 110, 113, 115, 121, 123, 130, 144, 149, 152, 160, 162, 165, 171, 174, 183, 188, 195, 200 |
| Long-term cut-off period, in days | 667, 672, 715, 675, 681, 684, 688, 690, 677, 697, 680, 700, 765, 755, 732, 729, 720, 769, 770, 800 |
| Number of harmonic terms | 2, 4, 6 |
| Degree of polynomial function | 1, 3, 5 |

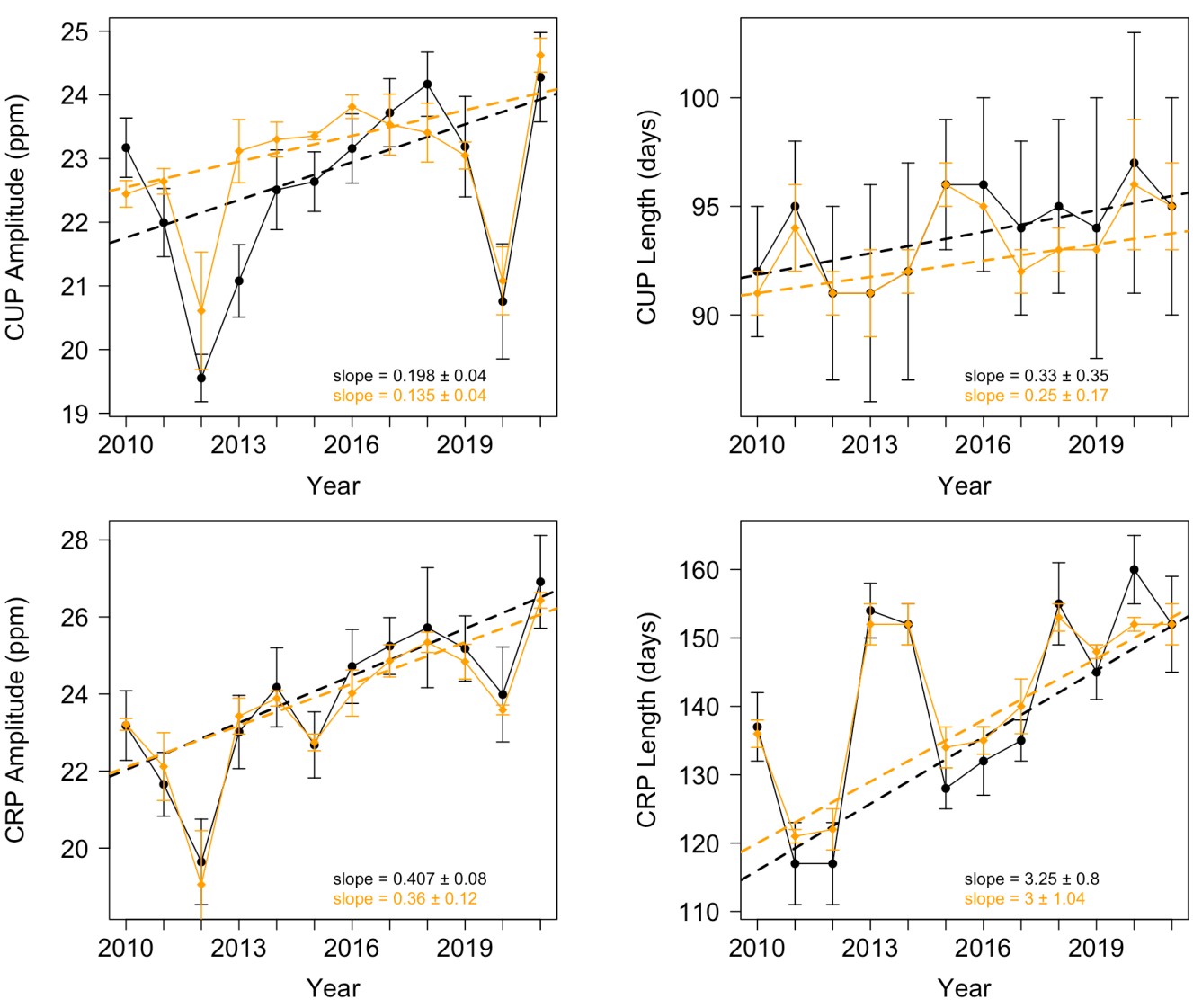

**Figure C1. Time series of the timing of CRP and CUP's onset and termination using two different curve fitting methods to smooth the atmospheric mole fraction data: Thoning et al. (1989) in black and HPspline in orange. The HPspline-derived smoothed concentration curve is stiffer than Thoning et al. (1989) and less sensitive to parameter settings.**

**Therefore, we created an ensemble of 4 extreme HPspline settings where the smoothing factor varies between 30, 500, 1000, 11000, and 99000.**

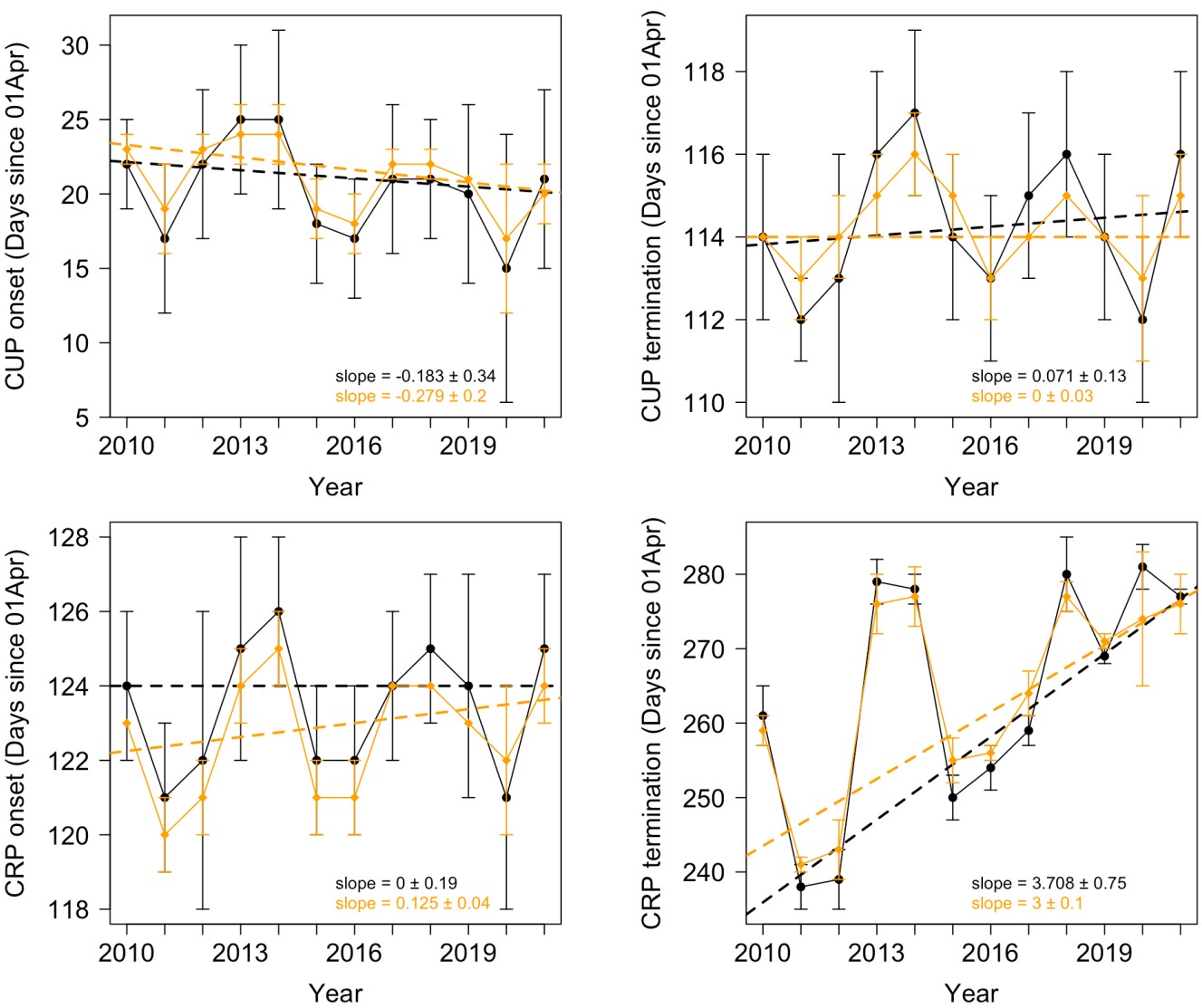

**Figure C2. Time series of the intensity of CRP and CUP (i.e., amplitude and length) using two different curve fitting methods to smooth the atmospheric mole fraction data: Thoning et al. (1989) in black and HPspline in orange. The**
**HPspline-derived smoothed concentration curve is stiffer than Thoning et al. (1989) and less sensitive to parameter**

settings. Therefore, we created an ensemble of 4 extreme HPspline settings where the smoothing factor varies between 30, 500, 1000, 11000, and 99000.

**Appendix D: Stations used in the global inversion.**

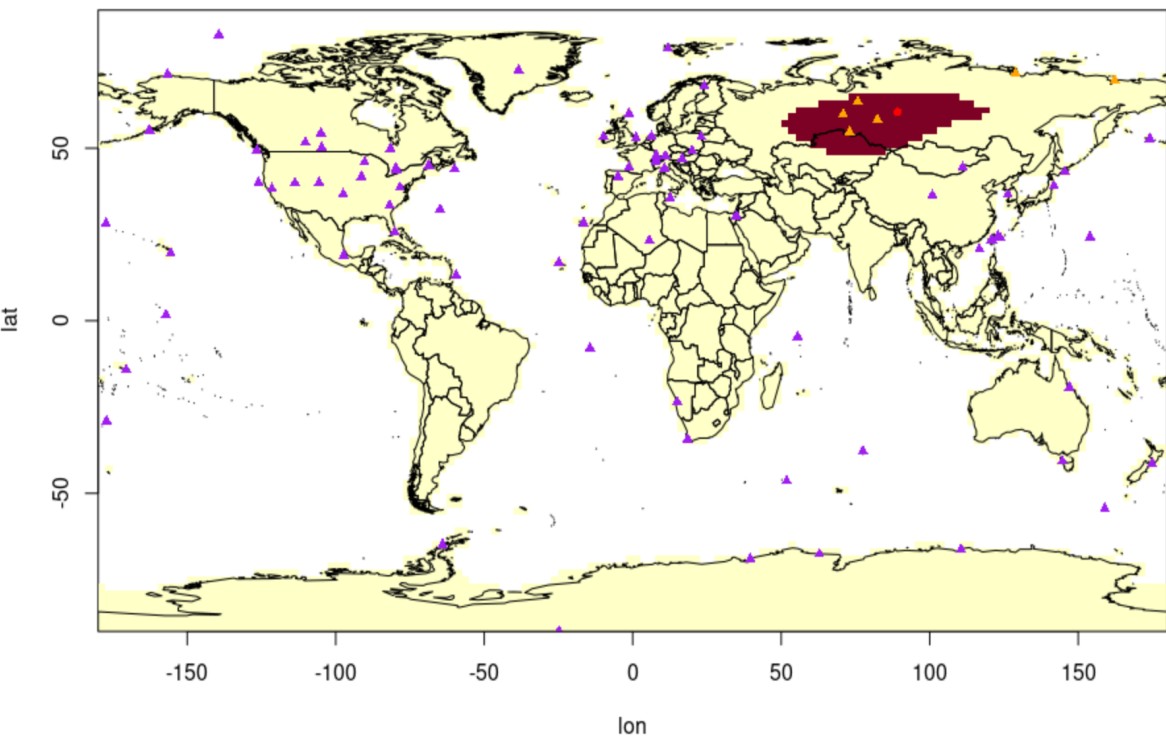

**Figure D1. Locations of the stations in the s10v2022 set (the purple triangles), ZOTTO (the red dot), and the additional six stations (the orange triangles) in the s10v2021+Allstations set. The red shading is the averaged influencing region on ZOTTO observational dataset.**

**Table D1. Atmospheric CO₂ measurement stations used in the inversion.**

Institutions are referenced as follows: AGH, University of Science and Technology, Poland; BGC, Max Planck Institute for Biogeochemistry, Germany (Thompson et al., 2009); CISRO, Commonwealth Scientific and Industrial Research Organisation (Francey et al., 2003); CNR-ISAC, Italian Air Force Meteorological Service, Institute of Atmospheric Sciences and Climate (Colombo et al., 1994); EC, Environment Canada (Worthy, 2003); EMPA, Swiss Federal Laboratories for Materials Science and Technology; FMI, Finnish Meteorological Institute (Kilkki et al., 2015); JMA, Japanese Meteorology Agency (Watanabe

et al., 2000); KMA, Korea Meteorological Administration (Cho et al., 2007); LSCE, Laboratoire des Sciences du Climat et de l'Environnement, France (Monfray et al., 1996); NIES, National Institute for Environmental Studies, Japan (Tohjima et al., 2008); NIPR, National Institute of Polar Research and Tohoku University, Japan (Morimoto et al., 2003); NOAA, National Oceanic and Atmospheric Administration/Earth System Research Laboratory, USA (Conway et al., 1994); RSE, Ricerca sul Sistema Energetico, Italy; RUG, Centre for Isotope Research, Rijksuniversiteit Groningen, The Netherlands; SAWS, South

African Weather Service (Labuschagne et al., 2003); SIO, Scripps Institution of Oceanography (Keeling et al., 2005; Manning and Keeling, 2006); UBA, Umweltbundesamt, Germany (Levin et al., 1995); UEA, University of East Anglia, UK. d: in situ, day-time selected; f: flask; h: in situ, all hours; n: in situ, night-time selected.

| Station Code | Institution | Record type | Lat | Lon | Height (a.s.l) | Used for | | |
|---|---|---|---|---|---|---|---|---|
| | | | | | | s10v2022 | s10v2022+ZOT | s10v2022+Allstations |
| BRW | SIO, NOAA | h,f | 71.32 | -156.61 | 12.5 | yes | yes | yes |
| SHIPIC AB | SIO | f | 82.85 | -139.35 | 0 | yes | yes | yes |
| LJO | SIO | f | 40 | -126 | 15 | yes | yes | yes |
| MLO | SIO | h,f | 19.53 | -155.58 | 3397 | yes | yes | yes |
| SPO | SIO | f | -89.98 | -24.8 | 2810 | yes | yes | yes |
| SHM | NOAA | f | 52.72 | 174.11 | 27 | yes | yes | yes |
| MID | NOAA | f | 28.21 | -177.37 | 10 | yes | yes | yes |
| MNM | JMA | d | 24.29 | 153.98 | 27 | yes | yes | yes |
| CHR | SIO, NOAA | f | 1.7 | -157.16 | 3.5 | yes | yes | yes |
| SMO | SIO, NOAA | h, f | -14.24 | -170.57 | 51 | yes | yes | yes |
| CBA | SIO, NOAA | f | 55.21 | -162.71 | 41 | yes | yes | yes |
| ETL | EC | d | 54.35 | -104.98 | 493 | yes | yes | yes |
| EST | EC | d | 51.67 | -110.21 | 707 | yes | yes | yes |
| ESP | CSIRO, EC | f, d | 49.38 | -126.54 | 27 | yes | yes | yes |
| BRA | EC | d | 50.2 | -104.71 | 595 | yes | yes | yes |
| FSD | EC | d | 49.88 | -81.57 | 250 | yes | yes | yes |
| EGB | EC | d | 44.22 | -79.7 | 251 | yes | yes | yes |
| DWN | EC | d | 43.78 | -79.47 | 198 | yes | yes | yes |
| WSA | EC | f, d | 43.93 | -60.01 | 5 | yes | yes | yes |
| LEF | NOAA | f | 45.93 | -90.26 | -396 | yes | yes | yes |
| UTA | NOAA | f | 39.9 | -113.72 | 1332 | yes | yes | yes |
| NWR | NOAA | f | 40.04 | -105.6 | 3526 | yes | yes | yes |
| SGP | NOAA | f | 36.71 | -97.49 | 356.5 | yes | yes | yes |
| AMT | NOAA | d | 45.03 | -68.68 | -107 | yes | yes | yes |
| WGC | NOAA | d | 38.26 | -121.49 | -483.5 | yes | yes | yes |
| WBI | NOAA | d | 41.72 | -91.35 | -379 | yes | yes | yes |
| SCT | NOAA | d | 33.41 | -81.83 | -305 | yes | yes | yes |
| SNP | NOAA | d | 38.62 | -78.35 | -17 | yes | yes | yes |
| KEY | NOAA | f | 25.67 | -80.18 | 4.5 | yes | yes | yes |

| MEX | NOAA | f | 18.98 | -97.31 | 4469 | yes | yes | yes |
|---|---|---|---|---|---|---|---|---|
| RPB | NOAA | f | 13.16 | -59.43 | 19 | yes | yes | yes |
| SUM | NOAA | f | 72.6 | -38.42 | 3214 | yes | yes | yes |
| BMW | NOAA | f | 32.26 | -64.88 | 46.5 | yes | yes | yes |
| IZO | AEMET | n | 28.31 | -16.5 | 2392 | yes | yes | yes |
| CVO | BGC | f | 16.86 | -24.87 | 10 | yes | yes | yes |
| ASC | NOAA | f | -7.97 | -14.4 | 88.5 | yes | yes | yes |
| ZEP | NOAA | f | 78.91 | 11.89 | 479 | yes | yes | yes |
| PAL | NOAA, FMI | f, d | 67.96 | 24.12 | 571 | yes | yes | yes |
| SIS | CSIRO, BGC | f | 59.97 | -1.26 | 26.5 | yes | yes | yes |
| MHD | NOAA | f | 53.32 | -9.81 | 19 | yes | yes | yes |
| WAO | UEA | d | 52.95 | 1.12 | -10 | yes | yes | yes |
| LUT | RUG | d | 53.4 | 6.35 | 61 | yes | yes | yes |
| BIK | BGC | f, d | 53.22 | 23.03 | -300 | yes | yes | yes |
| KAS | AGH-UST | n | 49.23 | 19.98 | 1989 | yes | yes | yes |
| HPB | NOAA | f | 47.8 | 11.02 | 965.5 | yes | yes | yes |
| SSL | UBA | n | 47.92 | 7.92 | 1205 | yes | yes | yes |
| HUN | NOAA | f | 46.95 | 16.64 | -96 | yes | yes | yes |
| JFJ | EMPA, BGC | f, n | 46.55 | 7.98 | 3577.5 | yes | yes | yes |
| PRS | RSE | n | 45.93 | 7.7 | 3480 | yes | yes | yes |
| CMN | CNR-ISAC | n | 44.18 | 10.69 | 2169 | yes | yes | yes |
| CIB | NOAA | f | 41.81 | -4.93 | 848.5 | yes | yes | yes |
| LMP | NOAA | f | 35.51 | 12.62 | 50 | yes | yes | yes |
| BIS | LSCE | d | 44.38 | -1.23 | 73 | yes | yes | yes |
| WIS | NOAA | f | 30.41 | 34.92 | 319 | yes | yes | yes |
| ASK | NOAA | f | 23.26 | 5.63 | 2715 | yes | yes | yes |
| NMB | NOAA | f | -23.57 | 15.02 | 461 | yes | yes | yes |
| CPT | NOAA, SAW | f, d | -34.35 | 18.49 | 260 | yes | yes | yes |
| UUM | NOAA | f | 44.45 | 111.1 | 1012 | yes | yes | yes |
| WLG | NOAA | f | 36.28 | 100.91 | 3852.5 | yes | yes | yes |
| AMY | NOAA, KMA | f, d | 36.54 | 126.33 | 107.5 | yes | yes | yes |
| COI | NIES | f | 43.15 | 145.5 | 45 | yes | yes | yes |
| RYO | JMA | d | 39.03 | 141.82 | 280 | yes | yes | yes |
| YON | JMA | d | 24.47 | 123.01 | 50 | yes | yes | yes |
| HAT | NIES | f | 24.05 | 123.8 | 10 | yes | yes | yes |
| LLN | NOAA | f | 23.47 | 120.87 | 2867 | yes | yes | yes |
| DSI | NOAA | f | 20.7 | 116.73 | 8 | yes | yes | yes |

| SEY | NOAA | f | -4.68 | 55.53 | 7 | yes | yes | yes |
|-----|------|---|-------|-------|---|-----|-----|-----|
| PSA | SIO, NOAA | f | -64.85 | -64.03 | 12.5 | yes | yes | yes |
| SYO | NIPR, NOAA | h | -69 | 39.58 | 29 | yes | yes | yes |
| CRZ | NOAA | f | -46.43 | 51.85 | 202 | yes | yes | yes |
| MAA | CSIRO | f | -67.62 | 62.87 | 42 | yes | yes | yes |
| AMS | LSCE | d | -37.8 | 77.54 | 55 | yes | yes | yes |
| CYA | CSIRO | f | -66.28 | 110.52 | 55 | yes | yes | yes |
| MQA | CSIRO | f | -54.48 | 158.97 | 13 | yes | yes | yes |
| BHD | SIO | f | -41.4 | 174.9 | 85 | yes | yes | yes |
| KER | SIO | f | -29.03 | -177.15 | 2 | yes | yes | yes |
| CFA | CSIRO | f | -19.28 | 147.06 | 5 | yes | yes | yes |
| CGO | CSIRO, NOAA | f | -40.68 | 144.68 | 130.5 | yes | yes | yes |
| DEM | NIES | d | 59.79 | 70.87 | 75 | | | yes |
| KRS | NIES | d | 58.25 | 82.42 | 50 | | | yes |
| AZV | NIES | d | 54.71 | 73.03 | 100 | | | yes |
| NOY | NIES | d | 63.43 | 75.78 | 100 | | | yes |
| ZOT | BGC | f, d | 60.8 | 89.35 | 301 | | yes | yes |
| TIK | NOAA | f | 71.6 | 128.89 | 20 | | | yes |
| AMB | BGC | d | 69.62 | 162.3 | | | | yes |

**Appendix E: CO flasks dataset.**

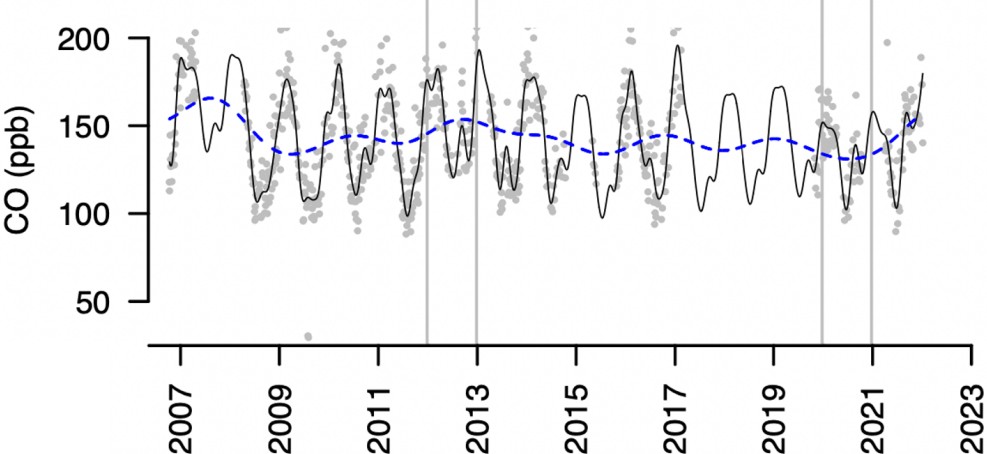


**Figure E1. Flask dataset (grey dots) for CO concentration from ZOTTO 301 m a.b.g. The black line is the smoothed curve for the dataset, and the dashed blue line is the running mean.**

**Appendix F: Timeseries of ZOTTO CO₂ mole fraction data.**

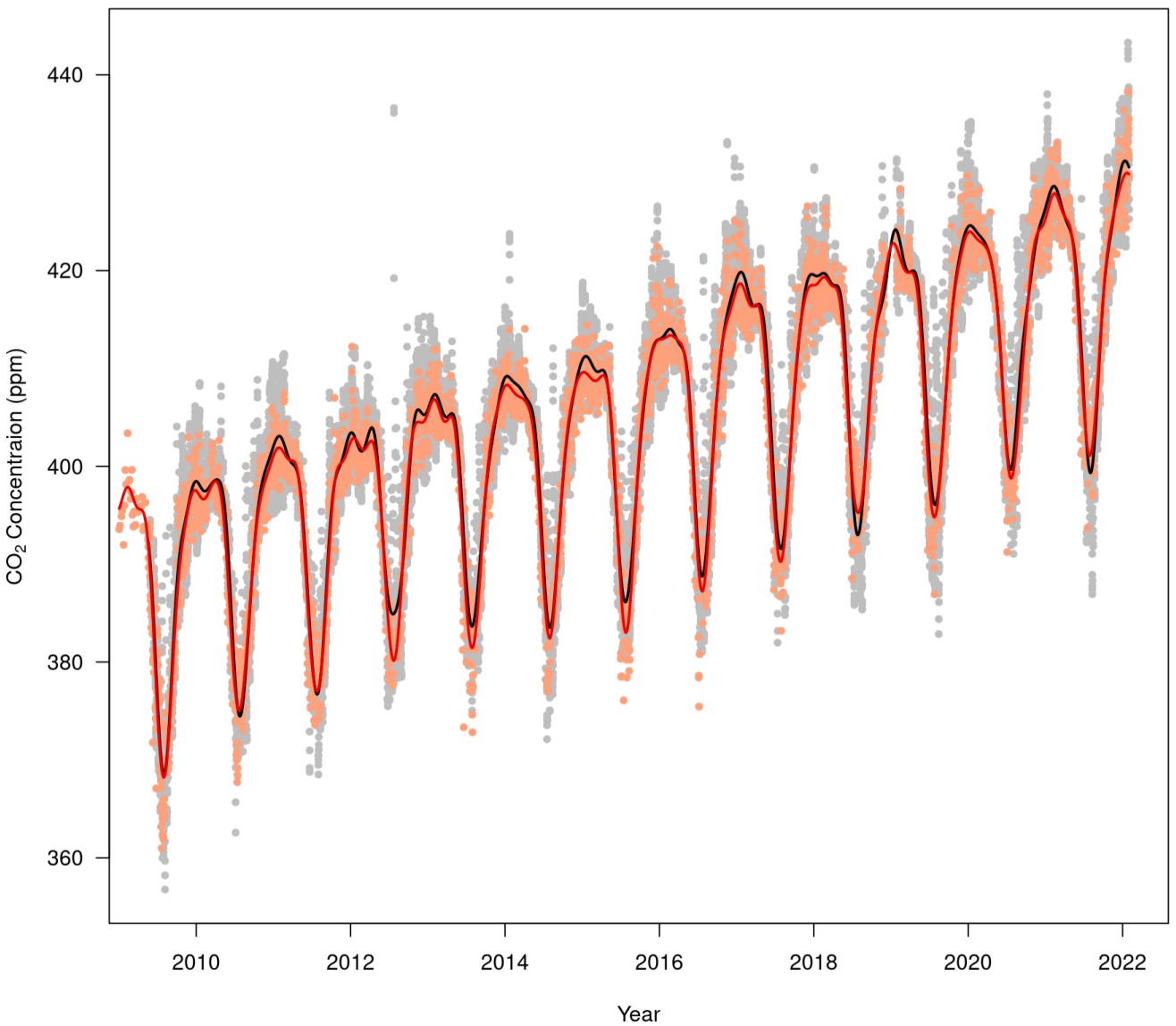

**Figure F1. Observed CO₂ mole fraction time series of ZOTTO (grey dots) and corresponding inversion forward simulation (s10v2022+ZOT) (light salmon dots); smoothed CO2 concentrations time series of ZOTTO observed data (black line) and inversion model forward outputs (red line) obtained from the Thoning et al. (1989) algorithm using short-term cut-off = 88 days, long-term cut-off = 667 days, number of harmonics = 2, and degree of polynomial = 3.**

**Appendix G: Analyses of inversions with different station sets listed in Table 1.**

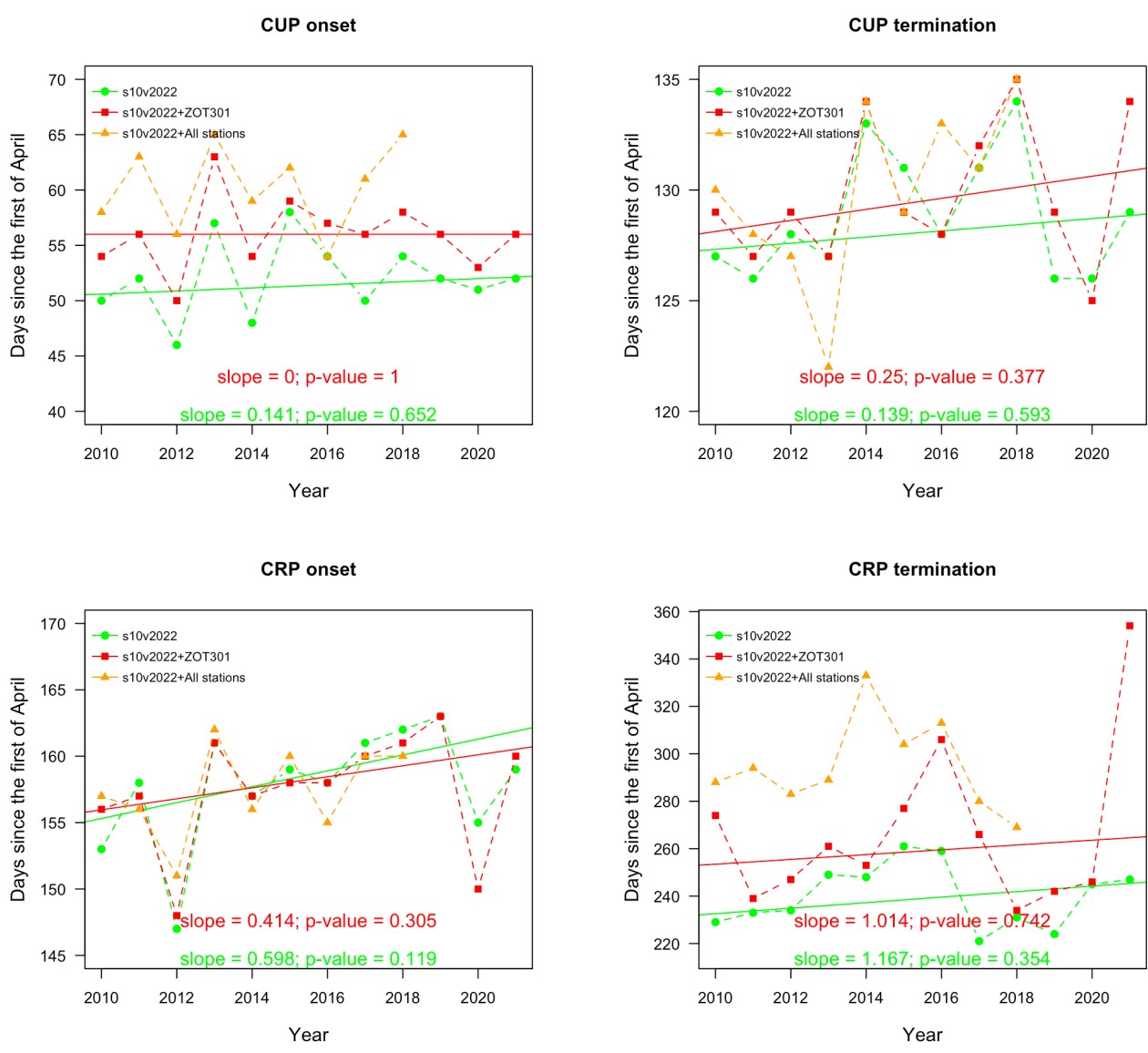

**Figure G1. Time series of the timing of CRP and CUP derived from three inversions with different station sets are listed in Table 1.**

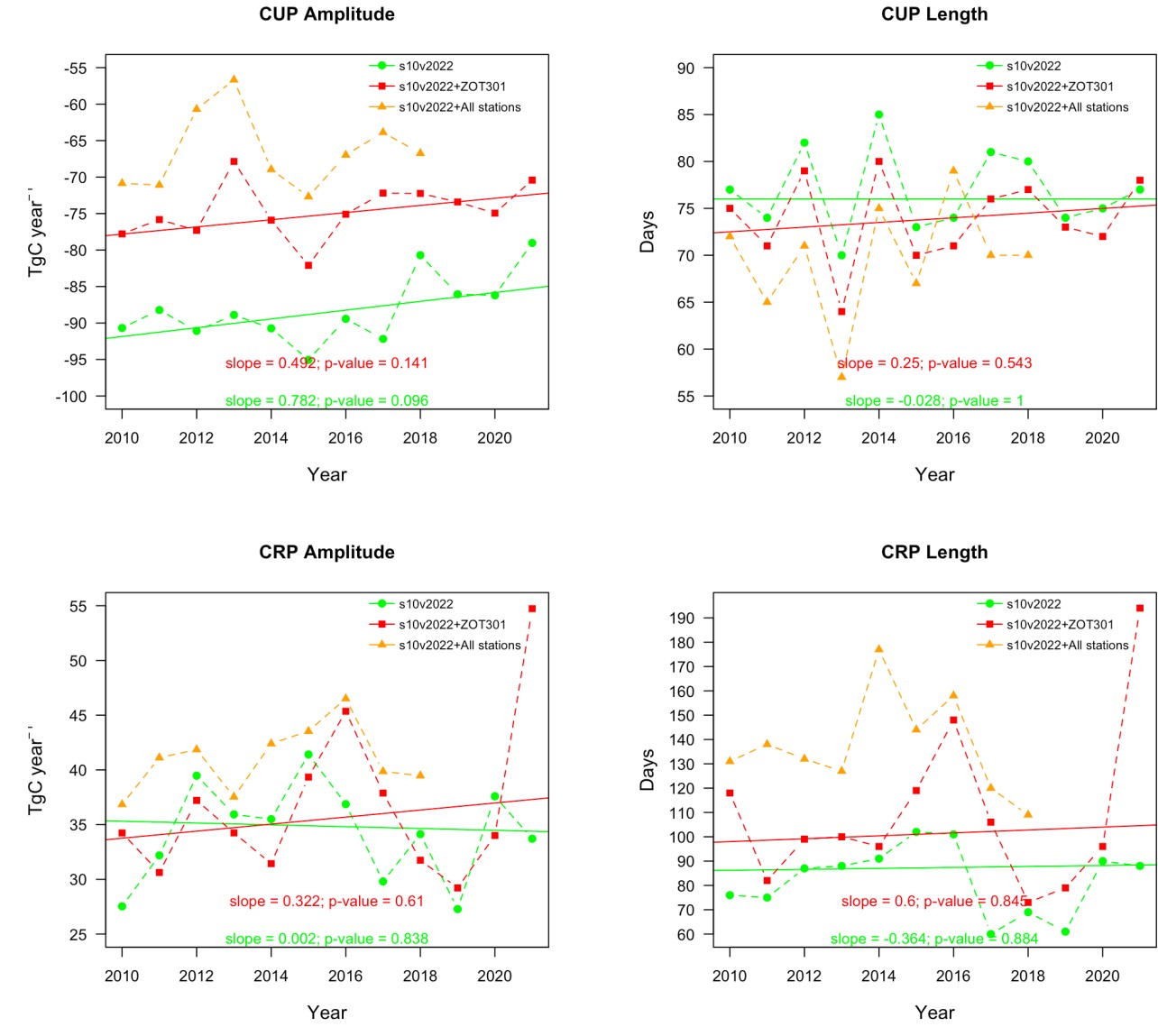


**Figure G2. Time series of the intensity of CRP and CUP intensity derived from three inversions with different station sets are listed in Table 1.**

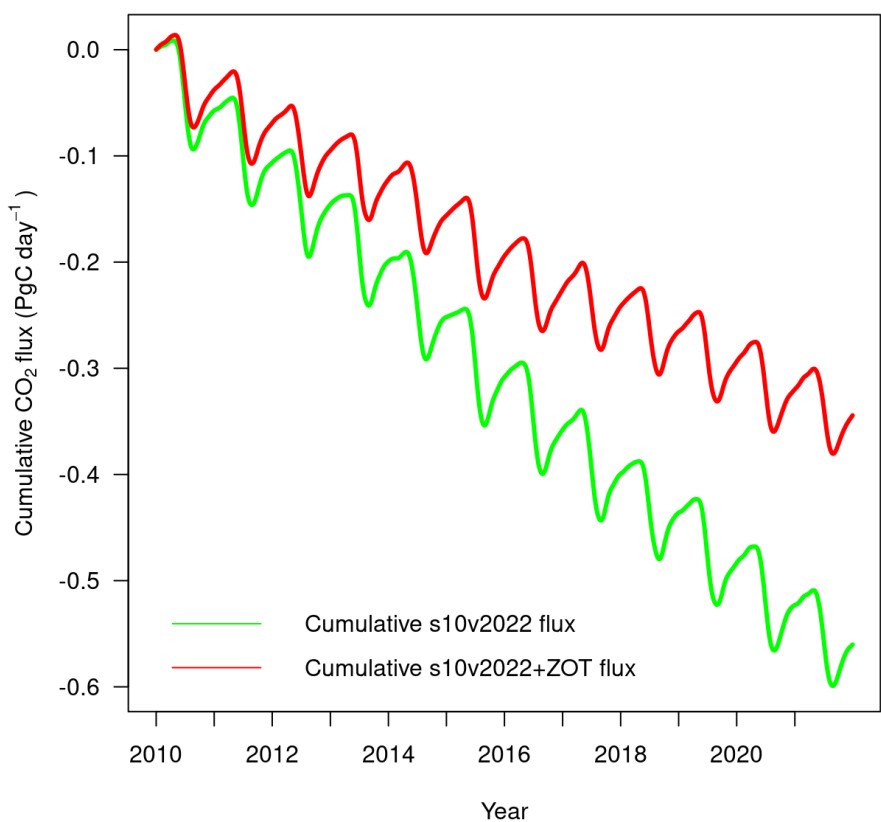


**Figure G3. Cumulative fluxes from two inversions using station sets s10v2022 and s10v2022+ZOT.**

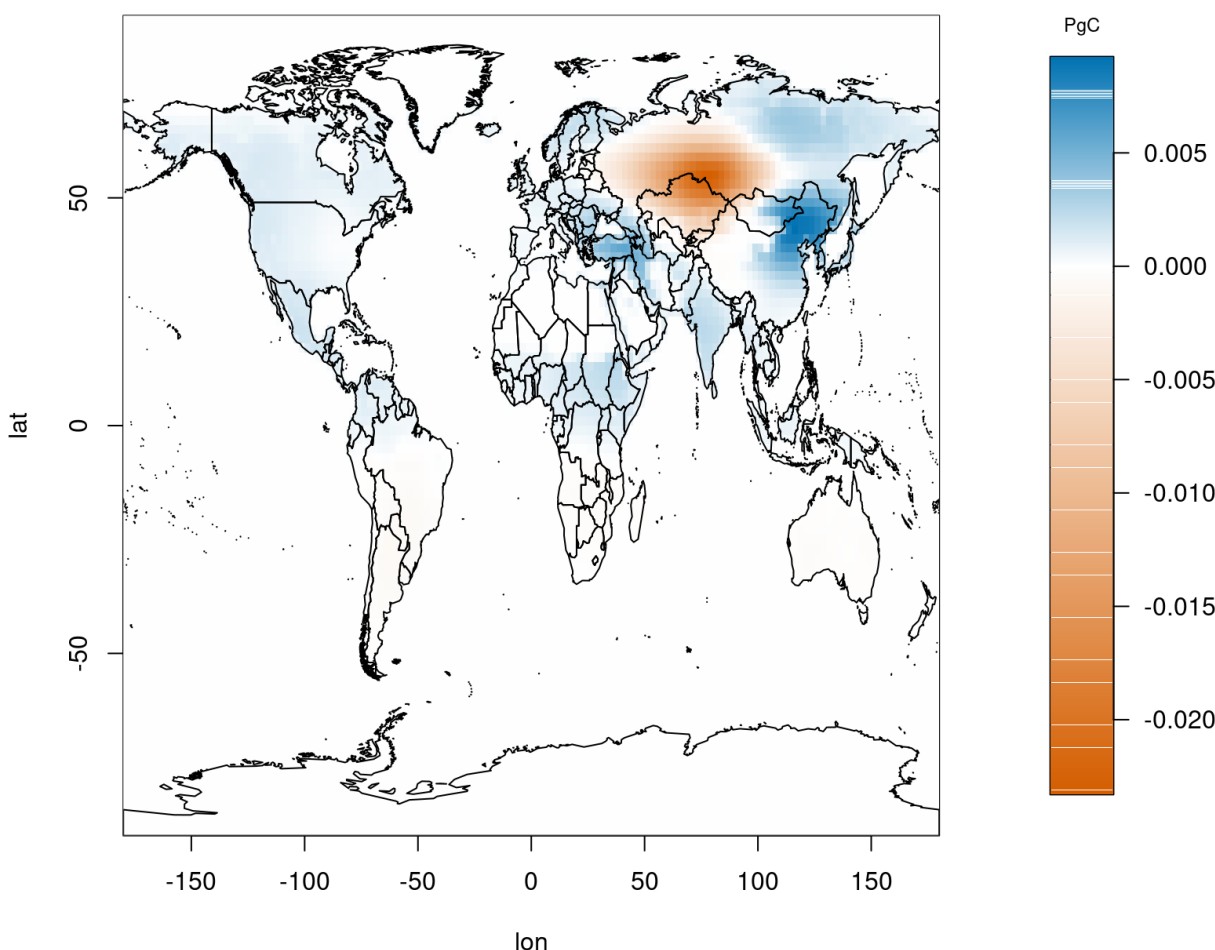

**Figure G4. Spatial differences in the 2010-2021 mean NEE between the inversions using station sets s10v2021 and s10v2021+ZOT.**

**Appendix H: Simulated forward mole fraction analysis:**

The simulated mole fraction data obtained from the inversion model underwent the same processing steps as the observed data, following the procedures described in sections 2.2 and 2.3 in the main text. The simulated mole fraction data has been calculated by forward simulation with the atmospheric tracer transport model TM3 driven with re-analysed meteorological data. Surface $CO_2$ fluxes supplied to the model are the inverse flux estimates based on atmospheric observations and the same transport model. By construction, therefore, the simulated atmospheric $CO_2$ mole fraction fields optimally fit the measurements at the set of observation sites used. In other words, the inversion has been constrained by exactly the same observations. Comparing the simulated mole fraction data to observations is essential to check how well inversion constrains the variabilities seen in the observations.

The trends in the intensity (i.e., length and amplitude) and the timing (i.e., onset and termination) of CUP and CRP derived from inversion forward mole fraction data are similar to those from observed data, despite smaller offsets in the absolute timings of onset and termination (Fig. H1 and H2). The interannual variations of CUP and CRP amplitude derived from observed data and model forward mole fraction are consistent. The $CO_2$ time series derived from the inversion captures the observed anomalies in the years 2012 and 2020. Indeed, when plotting observation-based analysis against forward output analysis (Fig. H3 and H4), the Theil-Sen linear fit slope between the two for CUP and CRP amplitude is 1.0 (R2 = 0.7) and 0.9 (R2 = 0.6), respectively. The inversion-based result also broadly captures the timing of CUP and CRP with few exceptions in the CRP termination date. This could be explained by the fact that the inversion-based mole fraction data has less short-term variability (Fig. F1) due to under-represented synoptic variability in the atmospheric transport model, and its smoothed mole fraction data obtained through the Thoning et al. (1989) method is also lower compared to that from the observed data. In general, the inversion is capable of reproducing internal variations and trends well. This gives us more confidence in the posterior NEE fluxes derived from the model that we will now use to further assess the signals and variations we have seen in the observation analyses. We also check the prior mole fraction data outputs of the inversion. The CUP and CRP calculated from the prior mole fraction data, which is the mean seasonal cycle of an inversion, contain very small year-to-year variations of the $CO_2$ flux, in particular no interannual variation similar to the posterior mole fraction analysis (Fig. H5 and H6).

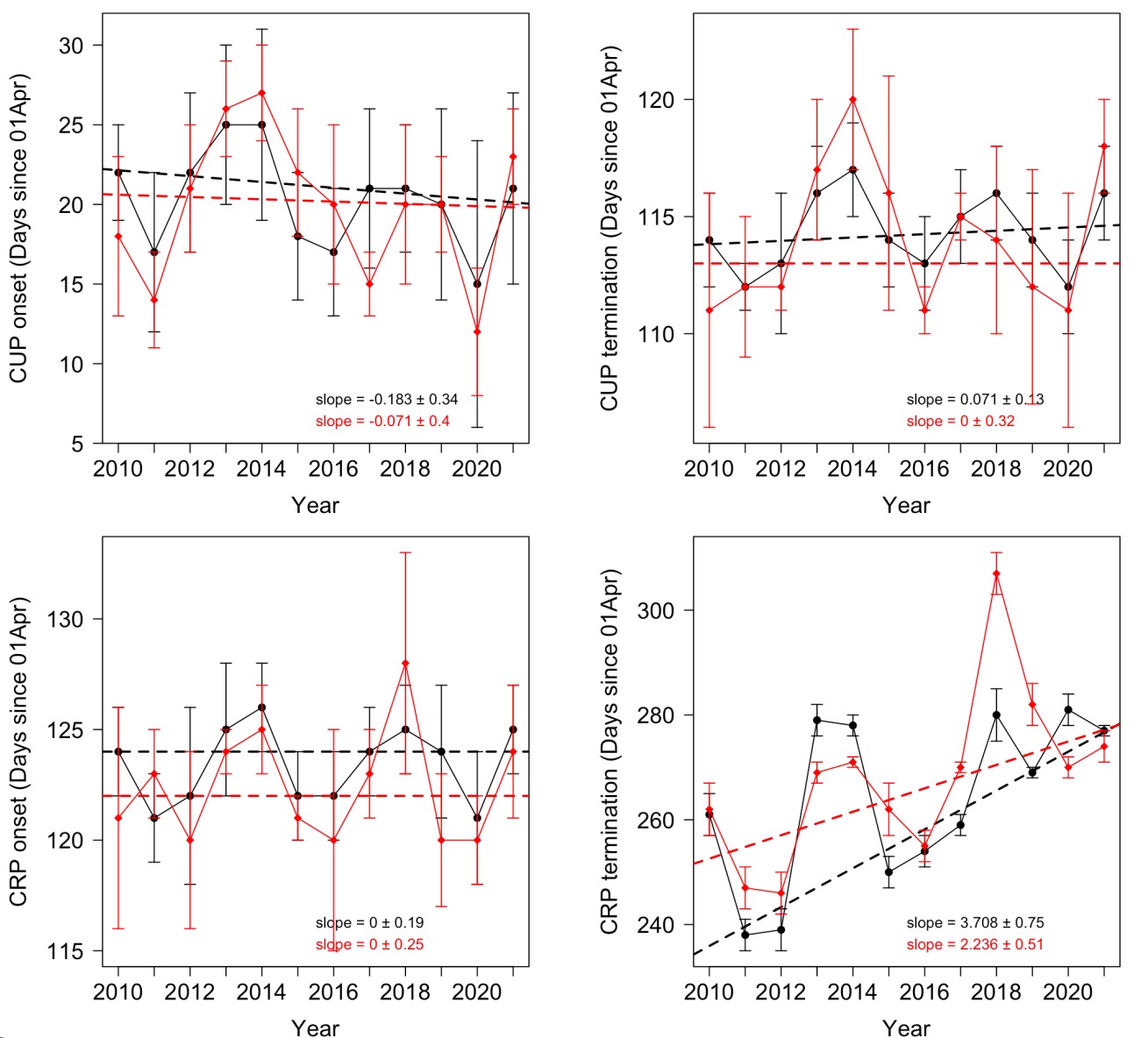

**Figure H1. Time series of the timing of CRP and CUP derived from observation (in black) and model forward (s10v2022+ZOT simulation) (in red) mole fraction analysis.**



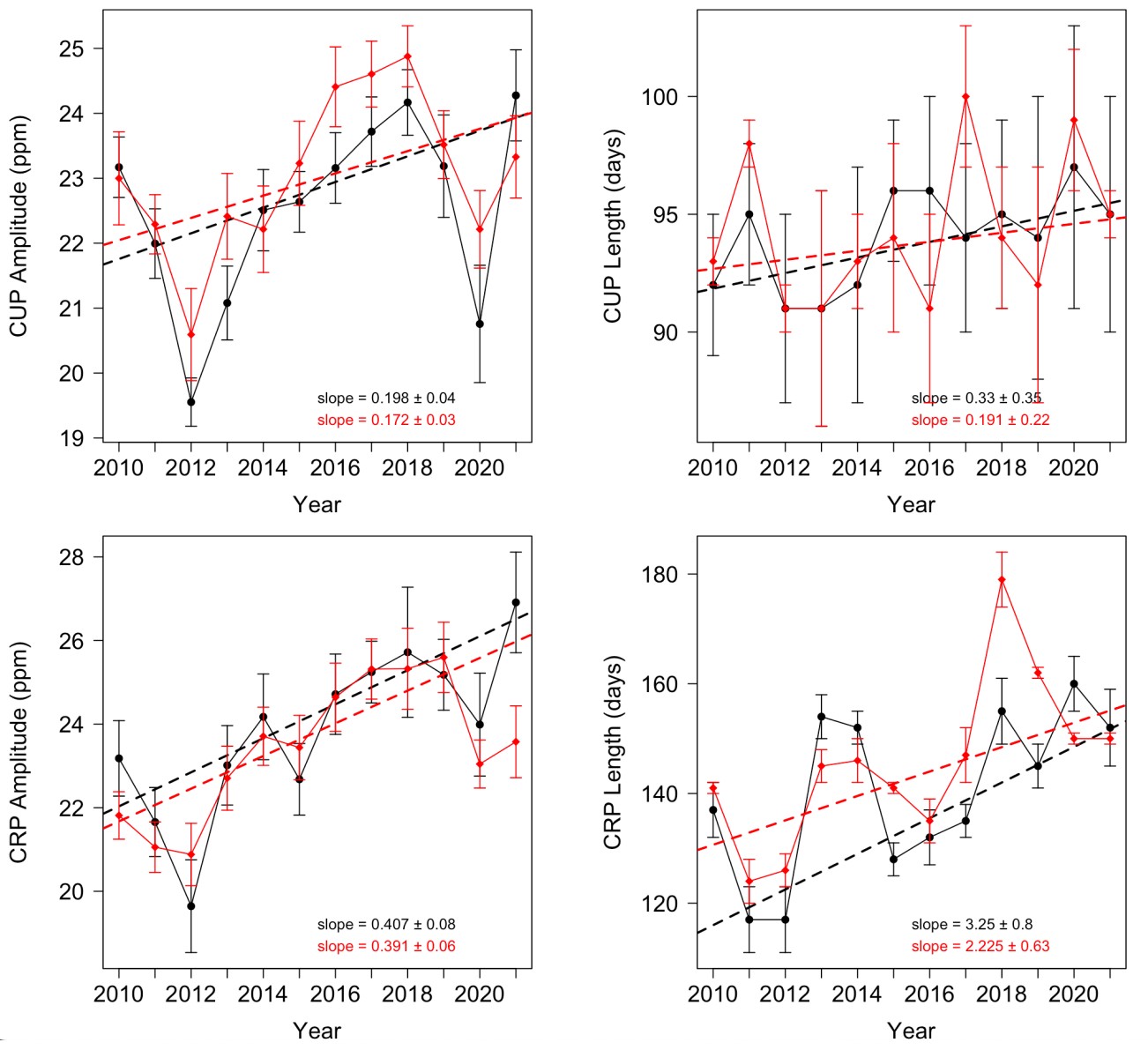

**Figure H2. Time series of the amplitudes and lengths of CRP and CUP derived from observation (in black) and model forward (s10v2022+ZOT simulation) (in red) mole fraction analysis**.

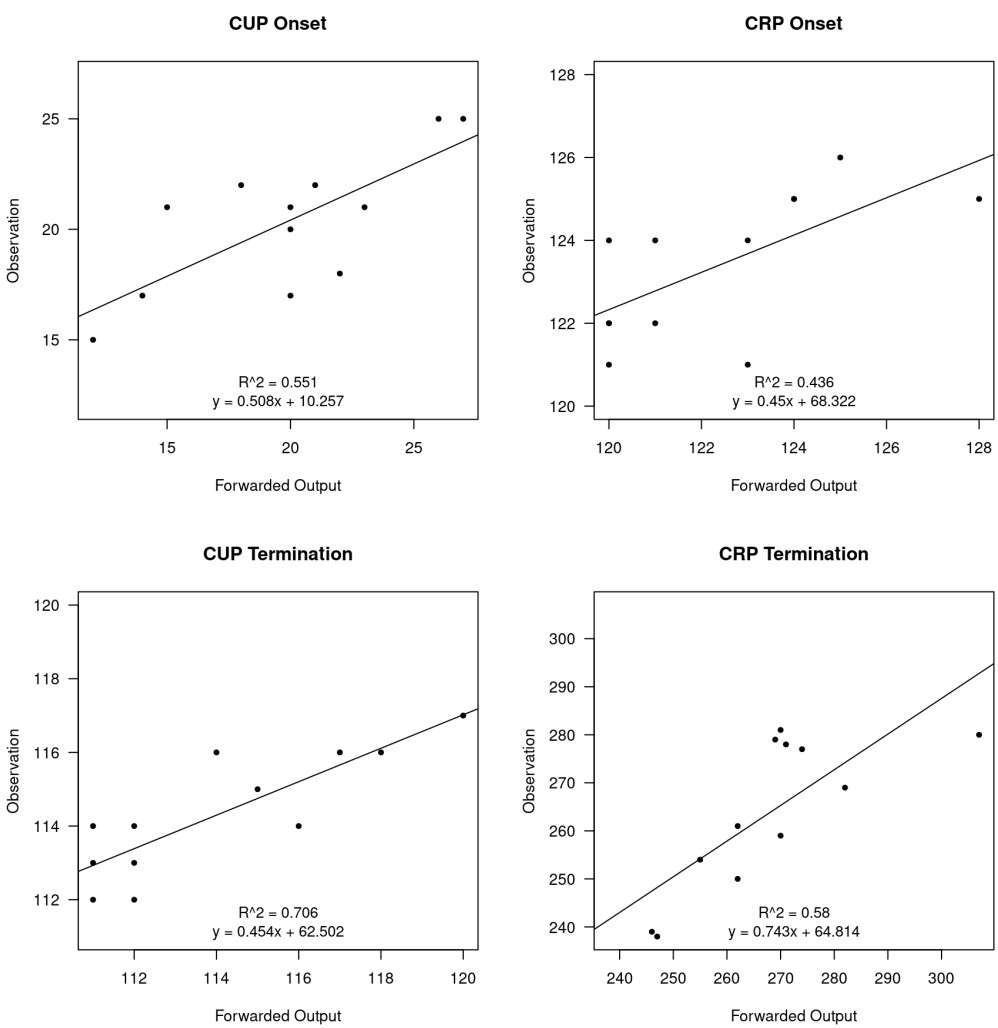

**Figure H3. Regression of the timing of CUP's and CRP's onset and termination derived from observational data against model forward output mole fraction.**

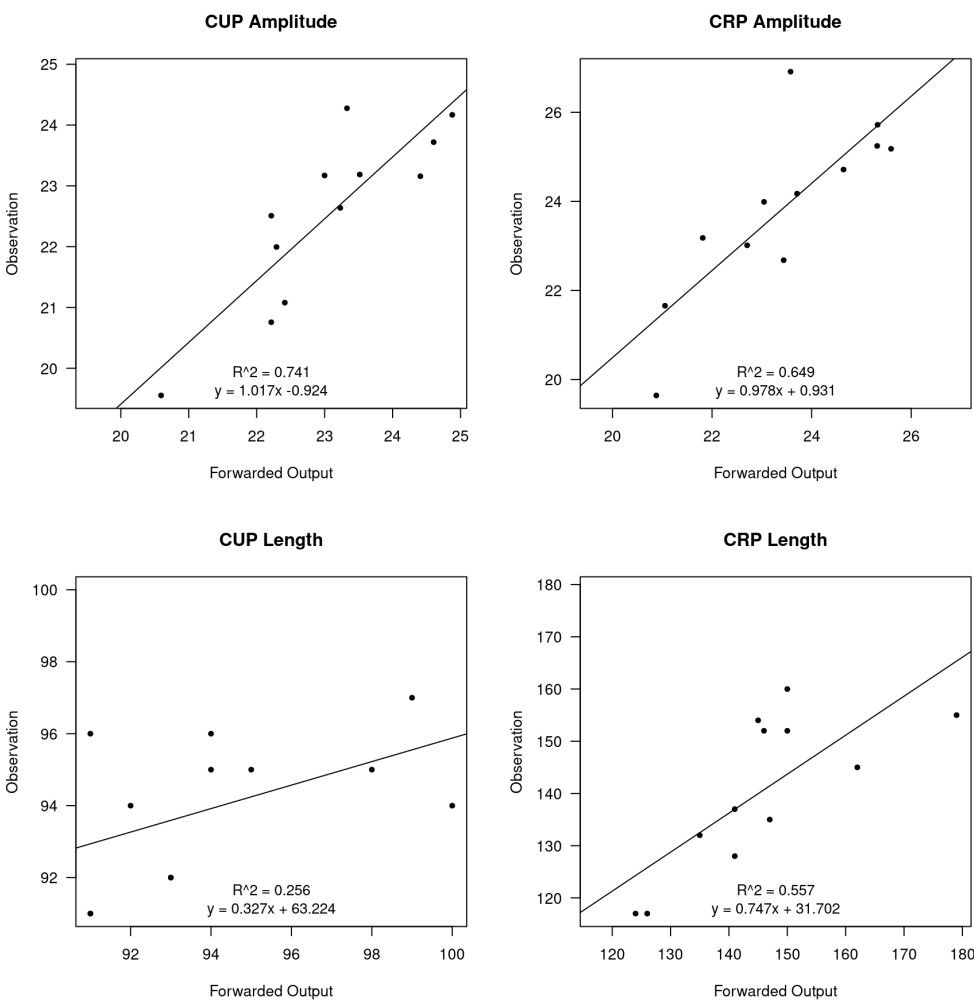

**Figure H4. Regression of intensity (i.e., amplitude of length) of CUP and CRP analyses derived from observational against from model forward output mole fraction.**

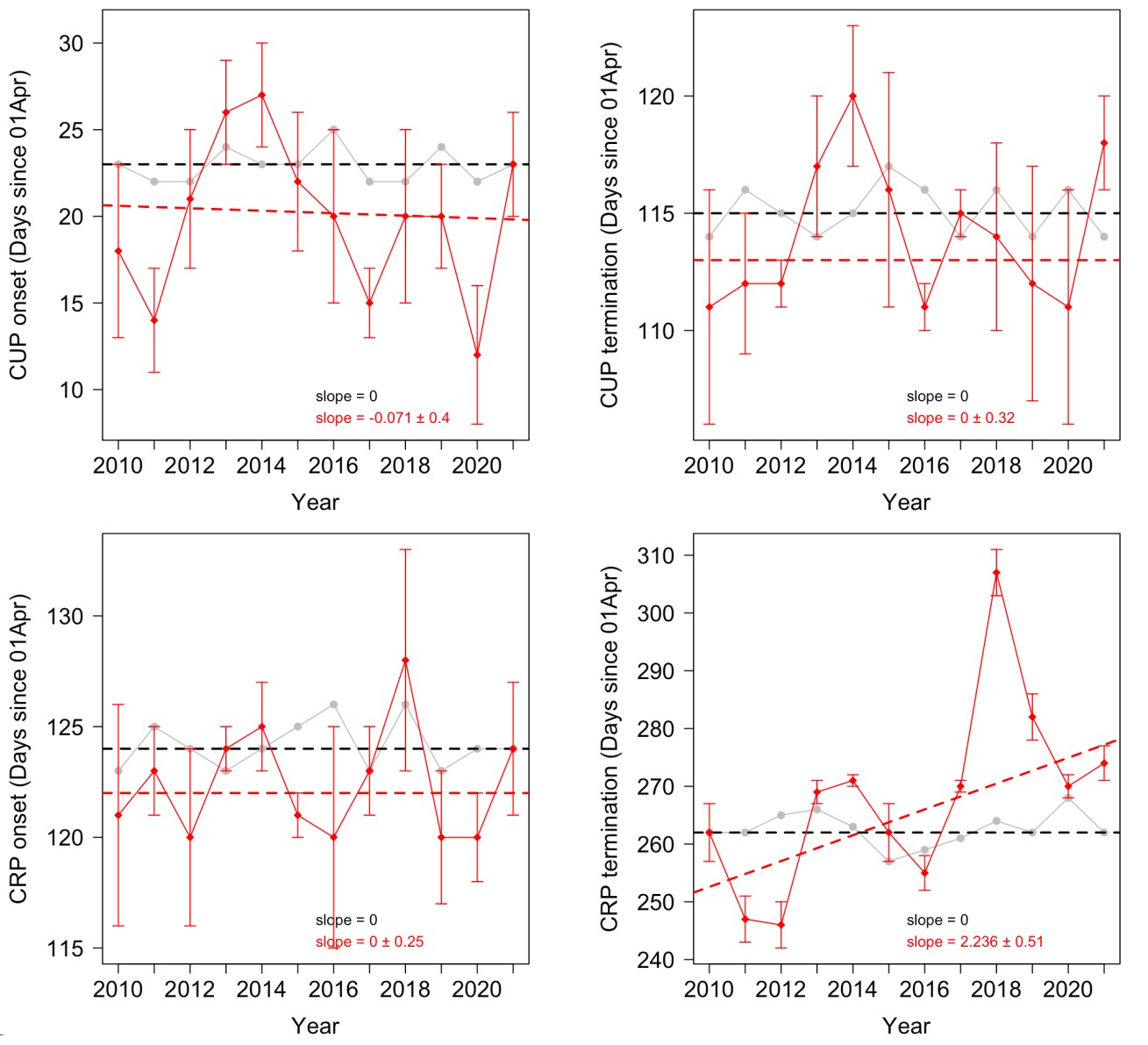

**Figure H5.** Time series of the timing of CRP and CUP derived from prior (in grey) and posterior (in red) model forward (s10v2022+ZOT simulation) mole fraction analysis.

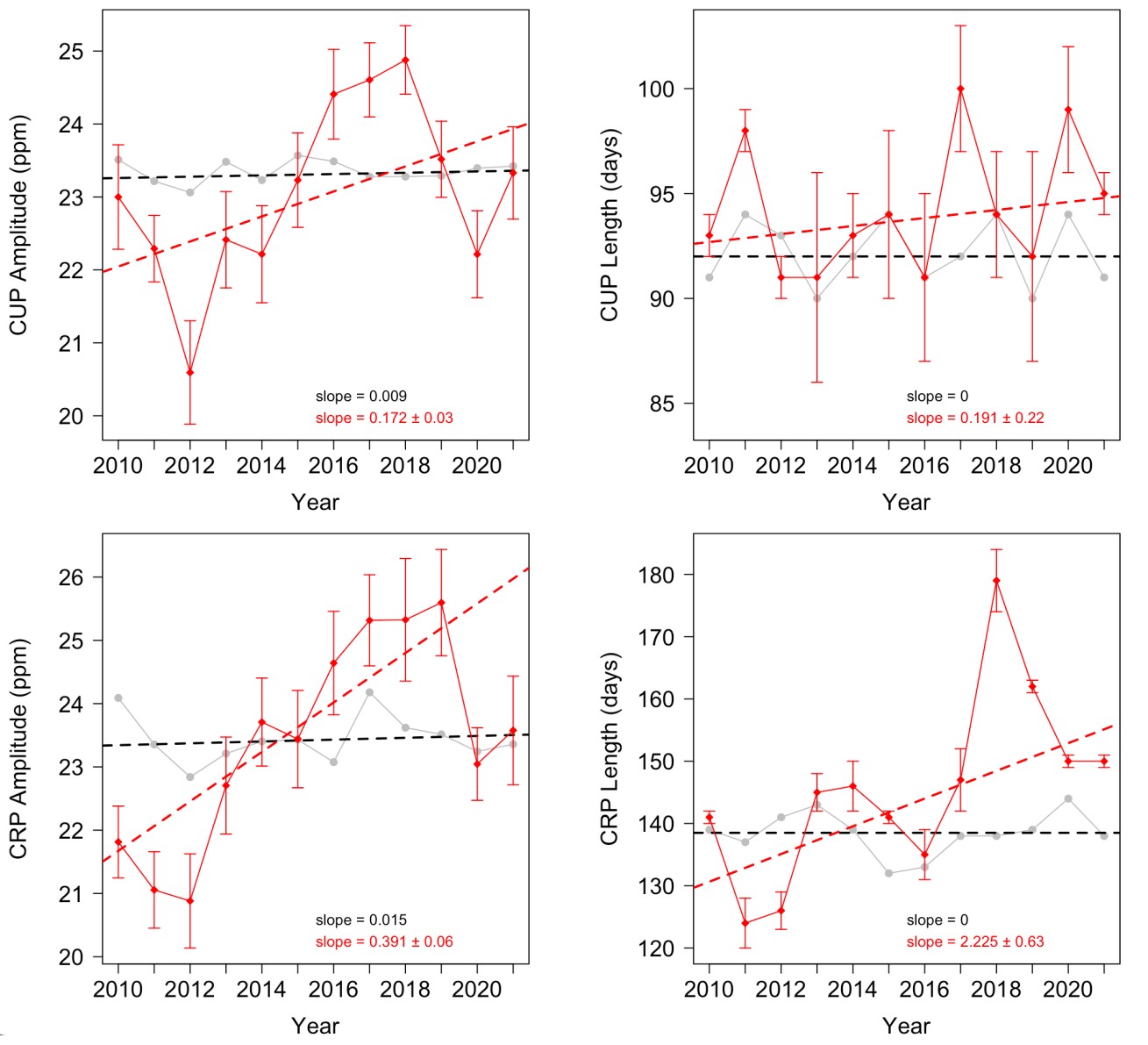

**Figure H6. Time series of the amplitudes and lengths of CRP and CUP derived from prior (in grey) and posterior (in red) model forward (s10v2022+ZOT simulation) mole fraction analysis**.

**Data Availability**

The $CO_2$ atmospheric mixing ratios are available on request at https://doi.org/10.17617/3.YBPFG2. This doi will be published upon the acceptation of the manuscript. More information can be given by Dieu Anh Tran (atran@bgc-jena.mpg.de). ZOTTO

$CO_2$ flask data is available at https://doi.org/10.17617/3.AXLVK0 (Jordan et al., 2023).

**Acknowledgement**

We acknowledge funding by the Max Planck Society to support the installation and maintenance of the ZOTTO until Feb. 2022. DAT acknowledges support from the International Max Planck Research School for Global Biogeochemical Cycles (IMPRS). For servicing the installed setup at the ZOTTO station we deeply appreciate the work of A. Panov, Anatoly

Prokushkin, and their colleagues from the Sukachev Institute of Forest in Krasnoyarsk, Russian Federation. Technical assistance to the upkeep of the instrumentation at MPI-BGC is also acknowledged, specifically J. Lavric, T. Seifert, S. Schmidt, U. Schultz, R. Leppert, and Karl Kübler.

**Author contribution**

DAT and SZ designed the study. DAT carried them out data curation and analysis. CG and CR advised on the use of the

CarboScope model and DAT performed and analysed the simulations. DAT prepared the manuscript with contributions from all co-authors.

**Competing interests**

At least one of the (co-)authors is a member of the editorial board of Atmospheric Chemistry and Physics.

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
