# Peer review of "Interannual Variations in Siberian Carbon Uptake and Carbon Release Period."

_EGUsphere, 2023_

## Author Comment (AC1)

**Respond to Reviewer #1**

This study presents an analysis of CO2 mole fraction data collected at the ZOTTO measurement site in Siberia. These data are an incredible resource for the community. The authors use a variety of statistical tools and an inverse method to interpret these mole fraction data and come to some conclusions – some that agree and some that appear to be at odds with previous studies. I have noted some weaknesses in their analysis that need to be addressed before I can support this study being accepted for publication. There are a number of grammatical errors that should be addressed before publication.

Thank you for the constructive review and suggestions. We detail our responses below.

Line 52: this list is not comprehensive. These authors appear to be leaning on CUP/CRP ideas presented elsewhere but not cited properly.

Thank you for pointing this out. We will add (Keeling et al. 1996; Pearman et al., 1981; Bacastow et al., 1985; Myneni et al., 1997) to the reference list of studies that used long-term CO2 records to monitor the dynamics of carbon exchange in northern ecosystems.

This reviewer notes that some of the primary references are used by Kariyathan et al, 2023. This is important because the primary references outline some of the issues raised by the present study. There is no theoretical way to remove a linear trend from any time series that describes a range of stationary and non-stationary processes – there will always be leakage between these variations irrespective of the method used to disentangle the individual signals. As a consequence of this, it is also difficult to directly split apart changes in maxima and minima associated with seasonal cycles. Although the authors have used error correlations to study some seasonal relationships. The CCGCRV method is a complicated but effective tool that involves fitting a series of harmonics but it cannot easily address non-stationary processes, which is most noticeable during rapidly varying environmental conditions that affect the amplitude and phase of the time series. This should at least be acknowledged somewhere, particularly because the authors study the influence of a heat wave and anomalous fire year.

We agree with the reviewer that the analysis of atmospheric time series is often a complex process because the data are usually highly autocorrelated and consist of periodic and irregular variations on both long and short timescales. However, as the reviewer mentions, while there is a need to apply filtering and curve-fitting techniques to obtain smooth and continuous data, there is no perfect way to remove a linear trend or diagnose changes in non-stationary periodic cycles such as seasonal cycles from any time series.

The first limitation of curve-fitting is that the application of a particular curve-fitting program in the analysis or decomposition of an atmospheric time series may introduce biases that could significantly influence the results and conclusions of an investigation (Nakazawa et al., 1997; Tans et al., 1989; Pickers and Manning, 2015; Barlow et al., 2015). Already in the previous version of the manuscript, we had created an ensemble of different parameter settings of CCGCRV as well as compared the influence of two different curve-fitting methods on our analyses. These results suggested that there were no significant differences in terms of trends and signals when we alternated different CCGCRV parameters as well as compared it with the HpSpline program.

The second limitation mentioned by the reviewer is that key aspects of the seasonal cycle of a CO2 time series are sensitive to different curve-fitting approaches since neither CCGCRV nor other existed harmonic-based curve-fitting methods can properly address non-stationary processes. In the revised version of the manuscript, we will acknowledge this limitation more clearly. However, we note that even considering this second limitation, of existing curve-fitting method, the smoothed time-series derived from CCGCRV for ZOTTO data still represented the influence of 2012 summer wild-fire well (lower CO2 concentration in summer 2012, Fig. 4b in the manuscript). This implies that our method can identify anomalous seasons, even if we have to acknowledge that the absolute magnitude of change may be somewhat different with a different curve-fitting method.

This reviewer is more than a little concerned with their method used by the authors to calculate their linear slopes, given the large year to year variations. Values will be disproportionately influenced by outliers. More robust methods include the Seigel or Theil-Sen estimators. These methods will provide a more rigorous assessment of any observed trends, particularly given the length and noisiness of the ZOTTO time series. As a consequence of using a simple linear regression, this reviewer is wondering whether the results presented will remain statistically inconsistent with previous studies. Certainly, eyeballing some of their figures it is hard to imagine assigning any non-zero trend. Depending on what they find, using a refined method to determine trends may influence results described later about correlations between seasonal temperature anomalies.

We agree with the reviewer's assessment and will change the results of the manuscript accordingly. We have applied the Theil-Sen methods to our trend analyses. As in the Fig R1. below, the trends and their p-values after applying the Theil-Sen method do not change that much from the previous version of the manuscript.

[Figure]

Figure R1. Trends and p-values of CUP and CRP amplitude, length, and rate after applying the Theil-Senn method, the grey colour results from the previous linear trend analysis.

Re inversions: these authors will be well aware that translating changes in atmospheric mole fractions to regional CO2 fluxes is complex, which in this case involves seasonal variations in atmospheric transport from lower latitudes. The imbalance between the size and distribution of observation networks at higher and lower latitudes may also render the posterior solution problematic. Is there a noticeable improvement in the model performance in describing the CUP and CRP metrics from the prior to the posterior fluxes?

The prior is the mean seasonal cycle of an inversion, such that the CUP and CRP calculated from it should not contain any year-to-year variations of the CO2 flux, in particular no interannual variation similar to the observational data. If the CUP and CRP diagnosed from atmospheric concentrations simulated by forward transport of the prior contain any interannual variations, then these would come from either transport variability or from inhomogeneous sampling. We run the inversion from the prior fluxes and do the same comparison as in Figs H2-3 in the Supplement material. We did not find any inter-annual variations in the prior analysis and will add an additional Figure in the supplement materials about this.

On a related note, this reviewer is not surprised by the influence of including this one site into a global inversion. Despite claims to the contrary, there are large gaps in our knowledge about the carbon cycle at high northern latitudes and across the tropics. It would be useful for this reader to understand what had to change elsewhere (via mass balance) due to this albeit small change in the regional carbon balance due to using ZOTTO data.

We understand that since the global carbon budget is closed at annual and longer timescales, adding ZOTTO data to the inversion affects regional uptake, with compensations in the rest of the world, dominantly at high northern latitudes and tropics as shown in Figure G4 in the manuscript. We pointed this out already in the previous version of the manuscript (lines 379-380): "This indicates that adding ZOTTO data shifted the estimated carbon uptake within the NH (Fig. G4)". To avoid misunderstanding, we will rephrase this sentence to "Since the global carbon budget is closed at interannual timescales, adding ZOTTO data to the inversion altered the estimated carbon uptake within the rest of the world accordingly to conserve mass, leading to higher carbon uptake spread widely across the NH tropical and mid-latitudes 20°-50° N."

Last point, re conclusions: Improved flux estimates will also come from satellite data, collected primarily during summer months when the observing geometry is favourable but also during other months via improved regional estimates of resolution transport model may improve regional estimates using sparse ground-based data, but whether the net impact is positive is debatable.

Thank you for pointing that out. We will expand our conclusions to note that due to the sparseness and uneven distribution of the monitoring surface networks, it is unclear whether a higher-resolution regional transport model alone may better constrain regional fluxes.

As the reviewer mentioned, satellite observations provide a more extensive and homogeneous coverage than in situ networks. Since they quantify a column-mixing ratio, and not only a local one like surface measurements, they provide a constraint on a larger fraction of the atmosphere than

surface observations. This is considered an advantage since it should lead to a more complete representation of the atmosphere, but it also makes inversions using satellite retrievals more sensitive to model errors in the upper troposphere and stratosphere (Monteil et al., 2013). Important drawbacks of satellite retrievals are that they are available only for a limited range of atmospheric conditions (absence of clouds, low aerosol load) and that they are less accurate and more difficult to validate than in situ measurements. Most importantly, since the satellite data are constrained to mainly summer months at high latitudes, their data are of limited value to constrain the full seasonal cycle in inversion models given the lack of constraint on high-latitude on the full seasonal cycle and its phasing. Therefore, it is still unclear whether flux estimated from satellite data, where ground-based observation is sparse, will provide an improvement from flux estimated from ground-based observation. We will include this consideration in the conclusion section of the manuscript.

**Minor points**

- Line 162: which 78 sites? Suggest they are listed somewhere.

The 78 sites were represented in Fig. D1 but additional table to a list of these stations will be provided in the supplement material of the revised manuscript.

- Line 263: 1958 to 1961?

The sentence "Our finding is consistent with Graven et al. (2013), using aircraft-based observations of $CO_2$ from 1958 to 1961 to show that the seasonal amplitude at altitudes of 3 to 6 km increased by 50% for high latitudes." Will be re-written as: "Our finding is consistent with Graven et al. (2013), comparing 2009-2011 aircraft-based observations of $CO_2$ above the North Pacific and Arctic Oceans to earlier data from 1958 to 1961 and found that the seasonal amplitude at altitudes of 3 to 6 km increased by 50% for high latitudes."

- Grammatical errors throughout. Worth a closer check by the authors when they revise their manuscript.

We agreed and will thoroughly address them in the revised manuscript.